# A KL-regularization Framework for Learning to Plan with Adaptive Priors

Álvaro Serra-Gomez [1]   Daniel Jarne Ornia [2]   Dhruva Tirumala [3]   Thomas Moerland [1]

## Abstract

Effective exploration remains a key challenge in model-based reinforcement learning (MBRL), especially in high-dimensional continuous control tasks where sample efficiency is critical. Recent work addresses this by using learned policies as proposal distributions for Model-Predictive Path Integral (MPPI) planning. Early approaches update the sampling policy independently of the planner, typically via deterministic policy gradients with entropy regularization. However, since the data distribution is induced by the MPPI planner, misalignment between the policy and planner degrades value estimation and long-term performance. To address this, recent methods explicitly align the policy with the planner by minimizing KL divergence to the planner distribution or by incorporating planner-guided regularization. In this work, we unify these approaches under the Policy Optimization–Model Predictive Control (PO-MPC) framework, a family of KL-regularized MBRL methods that treat the planner's action distribution as a prior in policy optimization. We show how existing methods emerge as special cases of this family and explore previously unstudied variants. Experiments demonstrate that these variants yield significant performance gains, advancing the state of the art in MPPI-based RL.

## 1. Introduction

Recent approaches to planning-enhanced MBRL such as TD-MPC (Hansen et al., 2022) have shown that effective planning can significantly improve performance in MBRL by refining a learned policy through trajectory optimization. In these methods, a learned policy and its associated (action) value function are used for trajectory sampling and

evaluation in a planning process (i.e. **sampling policy** and **bootstrap value function**). Then, the sampling policy is updated off-policy, relying on promising transitions provided by planning. This paradigm ensures that the planning policy continuously benefits from improvements in the learned sampling policy and bootstrap action value function, which supply increasingly promising samples and accurate evaluations to the planner.

A key limitation emerges when trajectories are evaluated under a bootstrap value function conditioned on states and actions unlikely to be visited by the planner. This distribution mismatch between the sampling and planning policies leads to unreliable bootstrap estimates and poor value function learning, especially for short horizons. Recent work addresses this by aligning the sampling policy with the planner via reverse KL minimization (Wang et al., 2025b), but is hindered by its reliance on partially outdated planning samples, which introduce variance into policy updates.

Despite differing formulations, emerging MPPI-based methods implicitly follow the same principle for interacting with the environment and updating the policy, revealing a growing but fragmented landscape. This motivates a unifying framework that clarifies commonalities, organizes design choices, and enables systematic extensions to push forward the state of the art.

The main contribution of this work is Policy Optimization–Model Predictive Control (PO-MPC), a novel general MBRL framework for MPPI-based approaches. PO-MPC builds on the TD-MPC2 world model, and casts the sampling policy learning step as an instance of KL-regularized RL, where the trajectory distribution induced by the learned sampling policy $\pi_{\theta_s}$ is regularized against an MPPI-induced prior $\pi_p$ with strength determined by a hyperparameter $\lambda$. In particular, our formulation enables:

- **Novel configurations.** We explore new algorithmic variants by tuning the KL-regularization strength $\lambda$.

- **Intermediate prior.** We introduce a learned prior, and show both theoretically and empirically that it shields $\pi_{\theta_s}$ from outdated planner samples in the replay buffer.

- **Flexible objectives for training the prior.** We demonstrate how alternative losses for training the MPPI-

---

[1]LIACS, Leiden University, Leiden, The Netherlands [2]University of Oxford, Oxford, United Kingdom [3]Google Deepmind, London, United Kingdom. Correspondence to: Álvaro Serra-Gomez <a.serra-gomez@liacs.leidenuniv.nl>.

*Proceedings of the 43rd International Conference on Machine Learning*, Seoul, South Korea. PMLR 306, 2026. Copyright 2026 by the author(s).

**Sampling Policy Loss**

$$J(\pi_{\theta_s}) = \mathbb{E}_{a \sim \pi_{\theta_s}} \left[ Q_{\widetilde{\theta}_Q}^{\pi_{\theta_s}, \lambda}(z_t, a_t) \right] - \lambda \, KL\left[\pi_{\theta_s}(\cdot \mid z_t) \| \pi_p(\cdot \mid z_t)\right]$$

$\lambda = 0$
**TD-MPC2**
(Hansen et al., 2024)

**PO-MPC (Ours)** ①
$\lambda \in [0, \infty)$

$\lambda \to \infty, \pi_p = \pi_p^{MPPI}$
**BMPC**
(Wang et al., 2025)

$\pi_p$

**Replay Buffer**

$\pi_p^{MPPI} = \pi_P^t$

②

③ **Learned**

$$\pi_p^{rkl} = \arg\min_\pi KL[\pi(\cdot \mid z_t) \| \pi_P^t(\cdot \mid z_t)]$$

$$\pi_p^{fkl} = \arg\min_\pi KL[\pi_P^t(\cdot \mid z_t) \| \pi(\cdot \mid z_t)]$$

*Figure 1.* Overview of the PO-MPC framework. **1)** Sampling-policy learning is formulated as a KL-regularized reinforcement-learning problem, controlled by the hyperparameter $\lambda$, where the learned sampling policy $\pi_{\theta_s}$ is regularized toward the action distribution computed by MPPI. **2)** Since querying the MPPI policy is computationally expensive, we either reuse previously stored samples $\pi_P^t$ or, as proposed in this work, learn an approximation of it (i.e., $\pi_p^{rkl}$ or $\pi_p^{fkl}$). **3)** Using different losses to learn this policy prior, as a proxy for the planner's policy, allows embedding distinct inductive properties into the resulting sampling policy $\pi_{\theta_s}$.

induced prior embed distinct properties in $\pi_{\theta_s}$, yielding superior performance.

We validate PO-MPC on challenging high-dimensional continuous control benchmarks. We recover recent methods in the state-of-the-art as limiting cases of the regularization strength (Hansen et al., 2024; Wang et al., 2025b), and demonstrate that intermediate values of $\lambda$ not only outperform both extremes in terms of sample efficiency and final performance, but also manage to learn in environments where these fail. These results highlight that a principled unification of MPPI-based approaches not only clarifies their design space but also drives concrete improvements in practice.

## 2. Related Work

**Model-based RL.** Model-based reinforcement learning (MBRL) (Moerland et al., 2023) studies the combination of model and policy learning in sequential decision-making problems. On the one hand, a learned model offers both extra data (Sutton, 1991) and/or allows planning and obtaining more informed actions (Silver et al., 2017) or value estimates (Feinberg et al., 2018). Conversely, learning offers an (approximate) solution over the entire input space that generalizes to unvisited state-actions (Ackley & Littman, 1989), which is indispensable to overcome the curse of dimensionality (Poggio et al., 2017).

**Planning and RL.** Our work builds on advancements in planning-based (and model-based) reinforcement learning (MBRL), particularly methods that leverage online planning to guide policy learning. In many such approaches, like TD-MPC and subsequent works (Hansen et al., 2022; 2024),

a learned policy provides initial actions for a trajectory optimizer or planner, which then refines these actions using a learned model. The optimized trajectories subsequently provide data for policy and value function updates. However, the policy update often relies only on the single best actions or resulting trajectories from the planner, discarding potentially valuable information about the broader action distribution explored during planning. Alternatively, (Zhou et al., 2025) proposes using diffusion generative models to create policy and dynamic model proposals, and use them to solve an MPC problem.

Other examples of RL enhanced planning include (Silver et al., 2017; Wang et al., 2025b), where a policy is learned by imitating a powerful planner (e.g., MCTS, MPPI). Other methods exploit other sources of demonstrations to bias RL policies towards more informed distributions (Bhaskar et al., 2024; Hu et al., 2023; Yin et al., 2022). While effective, these imitation or cloning approaches may constrain the learned policy to the planner's immediate behavioral vicinity, potentially limiting its ability to directly optimize the long-term task objective (action value function) beyond what the planner currently achieves. On the other end, recent planning algorithms make use of expert knowledge or pre-trained policies to better inform the planning action search, robustly adapting to changes in the reward/cost function (Trevisan & Alonso-Mora, 2024; Wang et al., 2025a). In contrast, PO-MPC differentiates itself by proposing to utilize the entire action distribution generated by the planner, not just sampled actions or trajectories, as a guiding prior for the RL algorithm to exploit synergies between RL policy synthesis and planning-based action improvement. Recently, adjacent works have been proposed that present particular instances of MPPI-based RL methods (Lin et al.,

2025; Hansen et al., 2022; Zhan et al., 2025). Their main principles and relationship to our framework are explicitly detailed in the Appendix F.

**RL as probabilistic inference.** The idea of using priors to guide exploration in RL has been considered in many forms, albeit largely in the model-free setting (Tirumala et al., 2022). Priors can be used to guide learning by creating a trust region to constrain the optimization procedure (Schulman et al., 2015; 2017; Wang et al., 2017; Abdolmaleki et al., 2018); as an expectation-maximization (EM) update (Peters et al., 2010; Toussaint & Storkey, 2006; Rawlik et al., 2013; Levine & Koltun, 2013; Abdolmaleki et al., 2018) or to constrain learning in the offline or batch-RL setting (Siegel et al., 2020; Wu et al., 2019; Jaques et al., 2019; Laroche et al., 2019; Wang et al., 2020; Peng et al., 2020). A fundamental idea behind these works is to consider RL as a form of probabilistic inference where the policy being learned can be viewed as a posterior distribution over a prior and an objective (typically the exponentiated action value or advantage function) as in Levine (2018). In this work, we leverage this idea to reuse the model-based planning policy to guide learning its own sampling policy.

## 3. Preliminaries

We consider a discrete-time sequential decision-making problem over a horizon $T$, modeled as a Markov Decision Process (MDP) $(\mathcal{S}, \mathcal{A}, p, r, \gamma)$, where $\mathcal{S}$ is the state space, $\mathcal{A}$ the action space, $p(s' \mid s, a)$ the transition probability (or deterministic mapping) from state $s$ to $s'$ under action $a$, $r(s, a)$ the immediate reward for taking action $a$ in state $s$, and $\gamma \in [0, 1)$ the discount factor. A policy $\pi(a \mid s)$ defines a distribution over actions given the current state, and the objective is to find $\pi$ maximizing the expected discounted return

$$J(\pi) = \mathbb{E}_{\substack{s_0 \sim \rho_0, a_t \sim \pi(\cdot \mid s_t) \\ s_{t+1} \sim p(\cdot \mid s_t, a_t)}} \left[ \sum_{t=0}^{T-1} \gamma^t r(s_t, a_t) \right], \quad (1)$$

where $\rho_0$ is the initial state distribution.

**Reinforcement Learning (RL).** does not assume direct knowledge of $p$ or $r$; instead, an RL agent collects trajectories $\tau = (s_0, a_0, s_1, a_1, \ldots)$ through interaction and uses methods such as policy gradients, actor–critic, or value-based updates to learn a parametric policy $\pi_\theta(a \mid s)$ that maximizes $J(\pi_\theta)$ via trial-and-error.

**Model Predictive Control (MPC).** assumes access to a (possibly learned) model $p(s_{t+1} \mid s_t, a_t)$ and cost $c(s, a) = -r(s, a)$. At each time step $t$, MPC solves a finite-horizon

optimization

$$\min_{a_{t:t+H-1}} \mathbb{E} \left[ \sum_{k=0}^{H-1} c(s_{t+k}, a_{t+k}) \right] \quad (2)$$

$$\text{s.t.} \quad s_{t+k+1} = p(s_{t+k}, a_{t+k}),$$

over horizon $H < T$, applies the first action $a_t$, and then "recedes the horizon" by re-solving at $t + 1$ with the updated state. This online re-planning allows MPC to correct for model errors and disturbances. Both RL and MPC are methods to solve sequential decision-making optimisation problems: RL hinges on *learning* a global policy from experience, while MPC focuses on *online optimization* using an explicit model. In the next section, we show how Model Predictive Path Integral (MPPI) planning unifies these perspectives and can be further improved by incorporating learned policy priors via RL.

**Model Predictive Path Integral Control.** is a sample-based approach to solving planning methods that makes use of the fact that optimal stochastic control problems can be solved with path integrals to iteratively refine the optimal action distribution. At each step, it samples trajectories under a stochastic control law, weights them by cumulative cost, and refines its control sequence—all without requiring gradients of either dynamics or cost. Let $c(s_t, a_t)$ be a running cost (or reward $r = -c$) and $H$ a finite planning horizon. Denote a nominal open-loop control sequence by $\bar{a}_{0:H-1} = (\bar{a}_0, \ldots, \bar{a}_{H-1})$.

In MPPI, we sample $M$ noisy action sequences $a_t^{(i)} = \bar{a}_t + \epsilon_t^{(i)}$, $\epsilon_t^{(i)} \sim \mathcal{N}(0, \sigma_t I)$, and simulate $s_{t+1}^{(i)} \sim p(s_{t+1} \mid s_t^{(i)}, a_t^{(i)})$. Each trajectory $\tau_i$ has an associated cost

$$S(\tau_i) = \sum_{t=0}^{H-1} c(s_t^{(i)}, a_t^{(i)}). \quad (3)$$

After selecting the K-top performing samples, the MPPI update follows from a path-integral (desirability) transform:

$$\bar{a}_t \leftarrow \bar{a}_t + \sum_{i=1}^{K} w_i \, \epsilon_t^{(i)}, \quad \sigma_t = \sqrt{\frac{\sum_{i=1}^{K} w_i \left( \epsilon_t^{(i)} \right)^2}{\sum_{i=1}^{K} w^i}} \quad (4)$$

where $w_i = \exp\left(-\frac{1}{\lambda} S(\tau_i)\right) / \sum_{j=1}^{K} \exp\left(-\frac{1}{\lambda} S(\tau_j)\right), \lambda > 0$ is the temperature parameter, and controls how much the importance sampling scheme weights the optimal cost trajectory versus the others. After a fixed number of iterations, the planning procedure is terminated and a trajectory is sampled from the final return-normalized distribution over action sequences. Planning is done at each decision step and only the first action is executed to produce a feedback policy. To warm-start optimization and speed convergence, the mean control sequence is initialized with the 1-step shifted

$\bar{a}_{init} = \bar{a}_{t+1}$ from the previous decision step. We will denote the **planning policy** obtained after a fixed number of MPPI iterations by $\mathcal{N}(\bar{a}_0, \sigma_0 I)$ and $\pi_P$ interchangeably.

**MPPI-based Reinforcement Learning.** Prior work in Model-based RL (Bhardwaj et al., 2021; Hansen et al., 2022) has successfully applied MPPI to high-dimensional control tasks (i.e. DeepMind Control Suite (Tassa et al., 2018), Humanoid Benchmark (Sferrazza et al., 2024)) by planning in a learned a model of the MDP $(\hat{\mathcal{S}}, \mathcal{A}, \hat{p}, \hat{r}, \gamma)$, that differs from the original by using a learned latent representation of the state space $z = h_{\theta_h}(s) \in \hat{\mathcal{S}}$, an approximate reward $\hat{r}(z, a) = r_{\theta_r}$ and transition dynamics $\hat{p} = p_{\theta_d}$ (Bhardwaj et al., 2021).

In MPPI, trajectories are usually sampled from a Gaussian policy often initialized with zero mean and pre-set maximum variance to cover the action space almost uniformly, which is updated through multiple iterations of MPPI. Recent work (Hansen et al., 2024; Wang et al., 2025b) biases this sampling distribution, augmenting it with trajectory samples produced with a **learned sampling policy:** $\pi_{\theta_s}$. Since planning is done over a finite horizon, the learned sampling policy is also used for learning a **bootstrap action value function** $Q_{\theta_Q}^{\pi_{\theta_s}}$ evaluated on the last state of every sampled trajectory, leading to the H-step estimate: $Q(z_0, a_{0:H}^{(i)}) = \sum_{t=0}^{H-1} \gamma^t r_{\theta_r}(z_t, a_t^{(i)}) + \gamma^H Q_{\theta_Q}^{\pi_{\theta_s}}(z_H, a_H^{(i)})$.

Note that, since samples come from two distributions that are initially distinct, one learned and another initialized with high variance to enhance exploration, the trajectory distribution is bi-modal. MPPI approximates a softmax posterior of the bi-modal distribution modulated by the normalized exponential of the estimated H-step value function returns. This process is reminiscent of epsilon-greedy policies, where high-return actions are taken with high probability, leaving some probability mass for exploration.

**Policy Mismatch.** Initial MPPI-based RL approaches learn the *sampling policy* independently of the planner's action distribution (Hansen et al., 2022; 2024). In this context, training is still heavily influenced by the MPPI, which is used to collect transition data, but the sampling policy $\pi_{\theta_s}$ is not constrained to remain close to it. This decoupling creates a distribution mismatch: the action value function is trained under states and actions explored by the MPPI, but the sampling policy update maximizes locally the action value function (through deterministic policy gradients with entropy regularization (Hansen et al., 2024)[1].). Unless constrained to remain close to the MPPI's distribution, it is unlikely that the policy's local optima will match or be close to the local optima found by the MPPI, especially

in high-dimensional environments. This is the root of policy mismatch, which causes the action value function to be evaluated in underexplored state-action couples when bootstrapping. This results in poor action-value estimation, which compromises the approximation of action-value targets and further degrades the performance of MPPI. We refer the reader to (Wang et al., 2025b) for more details on Policy Mismatch.

# 4. Method

The challenge of Policy Mismatch when learning the sampling policy has been addressed in the past by directly cloning the planning distribution via reverse KL minimization $\mathrm{KL}(\pi_{\theta_s}(\cdot|z_t) \,\|\, \pi_P(\cdot|z_t))$ (Wang et al., 2025b)[2].However, this approach still suffers from:

- **Fixed KL penalty**: cloning the planning policy may collapse the sampling policy towards a local minima prematurely.

- **High-variance targets**: even when alleviated through *lazy reanalyze* (Wang et al., 2025b), cloning uses stale planner statistics stored in the replay buffer that mix many planner versions, effectively turning a unimodal MPPI posterior into a time-varying Gaussian mixture.

We propose instead to unify prior approaches that rely on action value maximization or KL minimization under a single perspective: sampling policy learning as KL-regularized RL toward a *planner-induced prior*. This view makes explicit how design choices (i.e., trade-off between action-value maximization and KL minimization, planning policy representation) map to previous methods, establishing a generalised framework, and exposing new, previously unexplored configurations.

## 4.1. Policy Optimization - Model Predictive Control

Given these considerations, we propose PO-MPC: a MBRL generalizing RL framework based on MPPI. The general algorithm pseudocode for PO-MPC training is presented in Algorithm 1. Following TD-MPC2's world model, previous approaches share a learned neural network **sampling policy**, $\pi_{\theta_s}$, and the **bootstrap action value function** $Q_{\theta_Q}^{\pi_{\theta_s}}$, which are respectively used for biasing trajectory sampling and estimating the return of the trajectory beyond the horizon [3]. However, they all differ in how the learned sampling policy is updated. KL-regularized RL is a field of study that trains

---

[1]Although (Hansen et al., 2024) reports using SAC for updating the sampling policy, their public code omits the entropy term in action value function estimation.

[2]Although (Wang et al., 2025b) reports minimizing the forward KL divergence, their public code minimizes the reverse KL, which leads to notable performance differences as discussed in this paper.

[3]Details on the implementation of MPPI and training of the bootstrapping action-value function can be found in Appendix B.

a policy to maximize its action-value function while regularizing the policy by minimizing the reverse KL-divergence to a second policy prior $\pi_p$. This regularization effect is modulated through a hyperparameter $\lambda$. In the following, we explain the main features of PO-MPC, being summarized as: **1)** Learning the sampling policy via KL-regularized RL, **2)** using a learned intermediate prior to represent the planning policy, which **3)** can be trained through different losses.

**Sampling policy learning via KL-regularized RL.** Given a state encoder $z = h_{\theta_h}(s)$ and a policy prior $\pi_p$, our framework considers the following objective:

$$
J(\pi_{\theta_s}) = \mathbb{E}_{\substack{s_0 \sim \rho_0, a_t \sim \pi_{\theta_s}(\cdot|z_t) \\ s_{t+1} \sim p(\cdot|s_t,a_t)}} \left[ \sum_{t=0}^{T-1} \gamma^t \left[ r(z_t, a_t) \right. \right. \tag{5}
$$
$$
\left. \left. - \lambda \mathrm{KL}[\pi_{\theta_s}(\cdot \mid z_t) \parallel \pi_p(\cdot \mid z_t)]\right] \right],
$$

where KL represents the Kullback-Liebler (KL) divergence between the policy and a prior distribution. The overall goal is to approximate, through the learned policy $\pi_{\theta_s}(\cdot \mid z_t)$, the distribution of trajectories generated by the prior policy $\pi_p(\cdot \mid z_t)$ reweighted by their exponential expected return. This is especially useful when prior policies are known that are likely to come across high-return regions in the state space, thus providing a promising trust region to explore around. As detailed in (Levine, 2018) for uniform policy priors, the objective in Equation 5 turns into the following step-wise objective:

$$
J(\pi) = \mathbb{E}_{a \sim \pi_{\theta_s}}[Q_{\tilde{\theta}_Q}^{\pi_{\theta_s}, \lambda}(z_t, a_t)] \tag{6}
$$
$$
- \lambda \mathrm{KL}[\pi_{\theta_s}(\cdot \mid z_t) \parallel \pi_p(\cdot \mid z_t)],
$$

where $Q_{\tilde{\theta}_Q}^{\pi, \lambda}$ is the KL-regularized action value function, which accounts for the expected return and the reverse KL divergence between the learned and the prior policy accumulated until the end of the episode. Then, the recursive Bellman equation for $Q_{\tilde{\theta}_Q}^{\pi, \lambda}$ is:

$$
Q_{\tilde{\theta}_Q}^{\pi_{\theta_s}, \lambda}(z_t, a_t) = \mathbb{E}_{\substack{s_{t+1} \sim p(\cdot|s_t,a_t), \\ a \sim \pi_{\theta_s}(\cdot|z_{t+1})}} \left[ r(z_t, a_t) \right. \tag{7}
$$
$$
\left. + \gamma \left( Q_{\tilde{\theta}_Q}^{\pi_{\theta_s}, \lambda}(z_{t+1}, a) - \lambda \log\left( \frac{\pi_{\theta_s}(a \mid z_{t+1})}{\pi_p(a \mid z_{t+1})} \right) \right) \right]
$$

Note that $\lambda$ controls how close to the prior policy we want the sampling policy to be, which is enforced through Equations 6 and 7.

In this work, we focus on learning the sampling policy $\pi_{\theta_s}$, and using the planning policy $\pi_P$ for obtaining an adaptive prior $\pi_p$. We will also consider the case where

we will maximize an entropy regularized objective to enhance exploration: $J'(\pi) = J(\pi) + \alpha \mathcal{H}(\pi)$, often used in KL-regularized RL as seen in (Tirumala et al., 2022). We point the reader to Appendix E for additional proof on KL-regularized policy evaluation and improvement.

**Prior policy design.** Setting $\lambda = 0$ updates the policy exclusively through action value function maximization and entropy regularization, recovering the cost function of TD-MPC2 (Hansen et al., 2024). Meanwhile, minimizing only the reverse KL-divergence of the policy and the past planning policy distributions stored in the replay buffer (i.e. $\lambda = \infty$) recovers the BMPC cost (Wang et al., 2025b).

We remark that this latter use of the planning policy samples as the prior introduces variance in the policy updates. The planning policy statistics (mean and variance) sampled from the replay buffer depend on old, less trained versions of the sampling policy. Therefore, for a given state, all sampled planning distributions have different modes, unlike the unimodal distribution that would result from MPPI under the current sampling policy and bootstrap action value function. This challenge is already recognized in (Wang et al., 2025b), and partially alleviated by periodically updating a small subset of the planning statistics stored in the replay buffer.

We propose further decreasing the variance in the policy update by introducing an intermediate policy, an **adaptive prior** $\pi_{\theta_p}$, that approximates the planning policy $\pi_P$. The benefits of this choice are twofold: **1)** it shields the sampling policy updates from the variance introduced by old planning policy samples (see Appendix G), and **2)** It can be trained with losses beyond reverse KL divergence, providing flexibility in how the planning policy $\pi_P$ is represented and, in turn, how the sampling policy is guided.

As in prior methods, we can train this adaptive prior by either minimizing the reverse KL-divergence:

$$
J(\theta_p) = \mathbb{E}_{(s,\pi_P) \sim D}\left[ \mathrm{KL}[\pi_{\theta_p}(\cdot \mid z_t) \parallel \pi_P(\cdot \mid z_t)] \right], \tag{8}
$$

or, as a straightforward alternative, the forward KL divergence:

$$
J(\theta_p) = \mathbb{E}_{(s,\pi_P) \sim D}\left[ \mathrm{KL}[\pi_P(\cdot \mid z_t) \parallel \pi_{\theta_p}(\cdot \mid z_t)] \right]. \tag{9}
$$

Note that this choice comes with no loss of generality when the adaptive prior results from minimizing 8 (see Appendix G). Exclusively minimizing the reverse KL divergence between the learned sampling policy and the adaptive prior policy still recovers the policy update from (Wang et al., 2025b) since both sampling and adaptive prior policies are unimodal Gaussian distributions, and minimizing the reverse KL divergence imitates the latter exactly. Also note that choosing a prior that minimizes the reverse KL-divergence (Equation 8) will bias the sampling policy to-

**Algorithm 1** PO-MPC (Main): Plan → Infer → Regularize

---

**Inputs:** environment $\mathcal{M}$, simulated world model $\tilde{\mathcal{M}}$, MPPI planner, sampler policy $\pi_{\theta_s}$, value $Q_{\theta_Q}$, buffer $\mathcal{D}$, KL weight $\lambda$, (optional) entropy $\alpha$

1: **for** $t = 0, \dots$ **do**
2:    **Plan (policy-as-prior):**

- $\bar{a}_{t:t+H}, \sigma_{t:t+H} \leftarrow \text{MPPI}_{\tilde{\mathcal{M}}}(z_t | \pi_{\theta_s}, Q_{\theta_Q}^{\pi_{\theta_s}}, \bar{a}_{init})$

- $a_t \sim \pi_P(\cdot \mid z_t) := \mathcal{N}(\bar{a}_t, \sigma_t^2 \mathrm{I});$ Env. Step in $\mathcal{M}$

- Push $(s_t, a_t, r_t, s_{t+1}, \bar{a}_t, \sigma_t)$ to $\mathcal{D}$

3:    **Update model $\tilde{\mathcal{M}}$ and Distill (adaptive prior):** sample $\mathcal{B} \subset \mathcal{D}$; update $\theta_p$.
4:    **Regularize & Improve (RL):**

- Update $Q_{\theta_Q}$ and with TD targets under $\mathcal{M}$ using $\pi_{\theta_s}$.

- Update $Q_{\hat{\theta}_Q}^\lambda$ and with KL regularized TD targets under $\mathcal{M}$ using $\pi_{\theta_s}$.

- Update $\pi_{\theta_s}$ by maximizing:
  $\mathbb{E}_{\substack{s \sim \mathcal{B}, \\ a \sim \pi_{\theta_s}}} \left[ Q_{\hat{\theta}_Q}^\lambda(z, a) \right] - \lambda \, \text{KL}(\pi_{\theta_s} \| \pi_{\theta_p}) + \alpha \, \mathcal{H}(\pi_{\theta_s})$.

5: **end for**

---

wards distributions that match one of the modes of the planning policy distribution, accelerating convergence but hurting exploration. Meanwhile, choosing priors that minimize the forward KL-divergence (Equation 9) will bias the policy towards a Gaussian distribution that encompasses the support of all sampled planning distributions, thereby enhancing exploration but delaying convergence. Further details on training the adaptive prior are included in Appendix B.

**Method Summary.** PO-MPC provides a common view over previous methods while addressing two core challenges of MPPI-based RL: policy/planner mismatch and high-variance in stored planning samples. We do this by casting policy learning as KL-regularized RL toward a distilled, *adaptive* planner prior. Concretely, MPPI produces a planning policy, which we distill into $\pi_{\theta_p}$ (via reverse or forward KL) to remove replay-induced variance; we then update the sampling policy $\pi_{\theta_s}$ with the KL-regularized objective in Eqs. 6–7, balancing return maximization, proximity to the planner (through $\lambda$), and entropy for exploration. This Plan→Infer→Regularize loop aligns the value function's rollout distribution with both the learned policy and the planner, improving stability and sample efficiency. The framework subsumes prior methods as special cases ($\lambda{=}0$ recovers TD-MPC2; $\lambda{\to}\infty$ with reverse-KL distillation recovers Variant 3[4] of Wang et al. (2025b)) while enabling principled choice between fast mode-seeking convergence

---

[4]Variant 3 in Wang et al. (2025b) is identical to BMPC except for the fact that it learns the bootstrap action value function $Q^{\pi_{\theta_s}}$ of the sampling policy instead of the value function $V^{\pi_{\theta_s}}$.

and broader support-covering exploration.

## 5. Experiments

We evaluate different configurations of the proposed framework (PO-MPC) on 7 challenging and high-dimensional continuous control tasks from DeepMind Control Suite (Tassa et al., 2018) (Humanoid and Dog) and 14 tasks from HumanoidBench locomotion suite (Sferrazza et al., 2024). These tasks cover a diverse range of continuous control challenges, including sparse reward, locomotion with high-dimensional state and action space ($\mathcal{A} \in \mathbb{R}^{21}$, $\mathcal{A} \in \mathbb{R}^{38}$, and $\mathcal{A} \in \mathbb{R}^{61}$ respectively). Each experiment is run on a single NVIDIA A100 GPU, taking from 7h to 15h to train a policy for 1e6 time-steps. For reproducibility, our implementation is available at https://github.com/alvaro-serra/pompc.git.

**Baselines.** We empirically support the claims in this work by comparing design choices already taken under this framework in the literature, namely TD-MPC2 (Hansen et al., 2024) and BMPC (Wang et al., 2025b). We also explore variations of PO-MPC by studying the effect of intermediate values of $\lambda$, the inclusion of the intermediate policy $\pi_{\theta_p}$, and an alternative way to train the latter. Table 2 provides an overview of the tested configurations, including published works. Note that we compare against the strongest BMPC variant reported in (Wang et al., 2025b).

We evaluate PO-MPC under the same hyperparameters of TD-MPC2, except for those related to PO-MPC (see Appendix A). At training time, our algorithm follows the same structure as TD-MPC2, except for the extra KL-regularized Q-value function for updating the sampling policy, and, if included, the learned policy prior that clones the MPPI; both changes are reported in Figure 1 as numbers 1 and 2. In practice, we did not observe significant training duration changes between the baselines and our method. At inference time, our algorithm is computationally equivalent to TD-MPC2 and BMPC since both policy prior and KL-regularized Q-value function are no longer needed.

### 5.1. Results

The objective of this section is to test PO-MPC from three angles. First, we make an empirical study of the effects of prioritizing return maximization over KL divergence minimization by choosing different values for $\lambda$. Second, we verify that employing an intermediate policy prior does not hurt the performance of PO-MPC. Finally, we show an example of how differently trained policy priors may serve to embed different properties in the learned sampling policy.

**Trading off return and KL divergence optimization.** The parameter $\lambda$ regulates the trade-off between two competing objectives in the policy updates: maximizing episode re-

*Table 1.* Final average return across 7 high-dimensional control tasks from DMControl Suite (Tassa et al., 2018), and 14 from HumanoidBench (Sferrazza et al., 2024). Interquartile Mean (IQM) of 5 runs and 95% bootstrap CI. The last row for each environment suite is the Aggregate IQM with 95% stratified bootstrap CI. Learning curves from the same runs are reported in Appendix D. **Bold** is best. Higher is better.

| Task | TD-MPC2 | BMPC | Ours ($\lambda$=0.1) | Ours ($\lambda$=1) | Ours ($\lambda$=9) |
|---|---|---|---|---|---|
| Dog-stand | 979 [972, 981] | 989 [985, 994] | 991 [986, 995] | **992 [991, 994]** | 989 [985, 990] |
| Dog-trot | 911 [256, 923] | 939 [929, 964] | **957 [930, 963]** | 943 [939, 949] | 945 [910, 963] |
| Dog-walk | 958 [949, 963] | 963 [959, 969] | **973 [970, 979]** | 969 [962, 971] | 966 [956, 980] |
| Dog-run | 618 [541, 668] | **730 [699, 769]** | 707 [686, 729] | 721 [680, 760] | 728 [652, 764] |
| Humanoid-stand | 923 [883, 937] | 948 [938, 959] | 957 [952, 961] | 955 [944, 963] | **961 [955, 964]** |
| Humanoid-walk | 920 [877, 927] | 946 [943, 948] | 946 [941, 953] | **948 [947, 950]** | 946 [942, 953] |
| Humanoid-run | 496 [422, 514] | 531 [501, 556] | **579 [554, 611]** | 577 [524, 598] | 559 [543, 601] |
| **Aggregate IQM** | 886 [639, 953] | 936 [732, 968] | **940 [725, 974]** | 939 [730, 971] | 938 [728, 972] |
| H1hand-bal.-h. | 66 [63, 74] | 58 [43, 59] | 62 [56, 73] | 68 [67, 73] | **69 [68, 72]** |
| H1hand-bal.-s. | 66 [34, 89] | 87 [70, 143] | 69 [60, 75] | 363 [176, 447] | **690 [252, 766]** |
| H1hand-crawl | 946 [863, 981] | 966 [922, 983] | 978 [965, 982] | **984 [974, 986]** | 982 [978, 985] |
| H1hand-hurdle | 97 [76, 184] | 196 [166, 213] | **221 [173, 288]** | 218 [180, 256] | 203 [172, 237] |
| H1hand-maze | 275 [172, 321] | 350 [314, 358] | 234 [151, 286] | 347 [335, 353] | **354 [350, 388]** |
| H1hand-pole | 142 [48, 244] | 775 [530, 904] | 338 [196, 572] | **960 [880, 963]** | 908 [829, 959] |
| H1hand-run | 149 [10, 322] | 828 [622, 899] | 899 [895, 907] | **907 [905, 911]** | 884 [754, 909] |
| H1hand-sit-h. | 245 [19, 643] | 821 [684, 915] | 902 [571, 906] | 839 [640, 918] | **914 [911, 916]** |
| H1hand-sit-s. | 912 [870, 926] | 932 [925, 934] | 919 [911, 928] | 933 [930, 937] | **935 [928, 938]** |
| H1hand-slide | 174 [134, 272] | 314 [284, 474] | 451 [151, 548] | **733 [626, 871]** | 620 [489, 670] |
| H1hand-stair | 58 [28, 68] | 81 [53, 111] | 108 [99, 170] | 326 [232, 362] | **339 [138, 592]** |
| H1hand-stand | 886 [768, 916] | **932 [930, 933]** | 893 [833, 913] | 930 [908, 936] | 932 [910, 938] |
| H1hand-walk | 634 [496, 809] | 932 [930, 934] | 834 [650, 920] | 931 [849, 935] | **934 [544, 936]** |
| **Aggregate IQM** | 255 [103, 598] | 590 [261, 868] | 532 [210, 829] | 723 [417, 917] | **744 [438, 912]** |

turns and minimizing the KL divergence from the adaptive policy prior. Table 1, along with its training curves in Figures 4- 5 (see Appendix D), compares the baselines against PO-MPC evaluations with a policy prior learned according to Equation 8, under different values of $\lambda$. Specifically, we consider $\lambda = 0.1, 1$, and 9, which correspond to approximate prioritization of KL divergence minimization of 10%, 50%, and 90%, respectively.

Our results demonstrate that regulating the proximity between the sampling and planning policies matches and slightly outperforms the baselines in the lower-dimensional tasks from DMControl Suite while significantly boosting performance in the higher-dimensional tasks from HumanoidBench. In particular, in high-dimensional tasks, intermediate values of $\lambda$ often clearly outperform other methods (i.e., Balance S., Crawl, Pole, Run, Slide, Stair, and Walk), sometimes managing to learn in environments where the baselines fail (i.e., Balance S., Slide, and Stair). These results are maintained when taking the aggregated IQM across tasks for both DMControl and Humanoidbench.

There is a subset of environments, however, where the performance improvement is moderate. We hypothesize this

might be due to two reasons: Exploration-hard environments (i.e. Maze, Balance-hard) require either high generalization or hierarchical policies. In any case, these environments require such a high degree of exploration that none of the methods manage to find a solution that solves the task. In simple environments, policy mismatch might not be a problem to begin with. In that case, the policy obtained by maximizing the Q-value function would converge to a similar solution to the planner.

Overall, PO-MPC with intermediate $\lambda$ values shows the highest average return in most tasks, achieving clear superior results with respect to the state-of-the-art in higher-dimensional ones (HumanoidBench), especially when $\lambda$ is carefully tuned. We show that the framework's performance degrades when $\lambda$ is too low, e.g. $\lambda \leq 0.1$, due to policy mismatch, as well as in the limit , e.g. $\lambda \to \infty$ (BMPC), where premature mode collapse may hinder exploration. Note that $\lambda = 1$ works well across most tasks, with larger $\lambda$ often helping in high-dimensional environments.

**Policy prior: Learned Intermediate policy vs. Planning replay data.** Continuing our experiments in Humanoid-Bench, Figure 2 shows that, on average across tasks, using

a learned intermediate policy instead of using the Planning policy samples from the replay buffer matches the performance of the latter and, in some cases, surpasses it. In Appendix G, we provide a theoretical analysis suggesting that this effect arises from the reduction in variance that results from tracking the intermediate policy prior, which can be approximated exactly by the sampling policy, instead of the ensemble of unimodal Gaussian Planning policy samples from the replay buffer that are partially outdated.

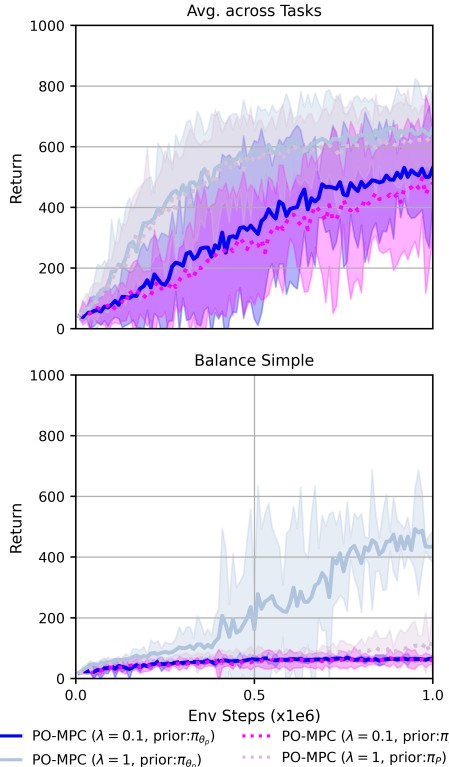

*Figure 2.* Effects of using a learned intermediate prior, $\pi_{\theta_P}$, instead of the Planning samples, $\pi_P$, from the replay buffer. Mean of 3 runs; shaded areas are 95% CI. We report the average across tasks **(Top)** and in the Balance Simple task **(Bottom)**. See Appendix D for results on all tasks.

**Policy prior Training.** Figure 3 exemplifies how choosing an alternative policy prior training cost changes the effect of different $\lambda$ values. For example, choosing priors minimizing the forward KL-divergence (Equation 9) will bias the policy towards a Gaussian distribution that includes the support of all sampled planning distributions, instead of matching the most frequent mode in the batch. This enhances exploration but delays convergence. This is why it is beneficial in environments where exploration is key, converging to a more stable solution faster at low values of $\lambda$ (i.e., in Stair); but detrimental in environments where deterministic behavior is crucial to obtain high rewards (i.e. Balance S.).

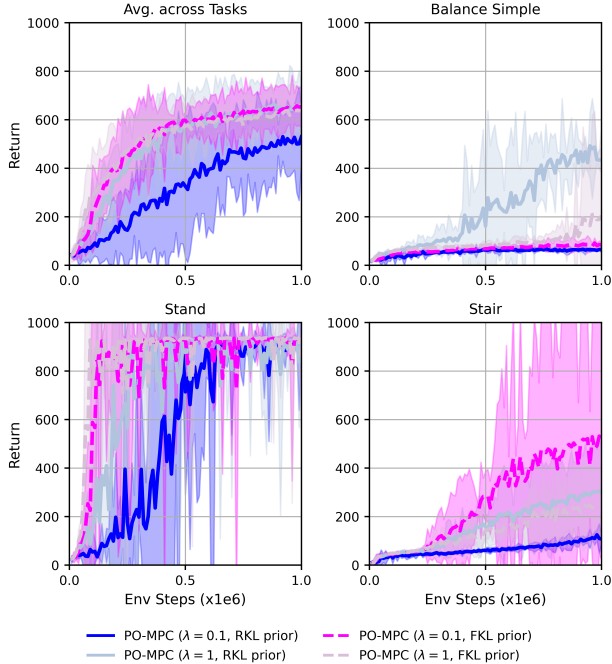

*Figure 3.* Effects of approximating the Planning policy with the intermediate prior through different cost functions. Mean of 3 runs; shaded areas are 95% CI. We report the average across tasks, and environments showing a clear effect of training with loss in Eq. 9 instead of Eq. 8. See Appendix D for results on all tasks.

## 6. Discussion and Conclusion

**Summary of Findings** Across 7 DMControl (Humanoid/Dog) and 14 HumanoidBench tasks, PO-MPC consistently improves over TD-MPC2 and BMPC. Table 1 shows that even modest KL regularization (e.g., $\lambda = 0.1$) yields sizable gains over TD-MPC2, with larger $\lambda$ often dominating in high-dimensional settings. Replacing on-replay planner samples with a learned *adaptive prior* matches or surpasses cloning-from-replay (Fig. 2), suggesting reduced update variance and smoother training (see Appendix G). The choice of prior fitting objective is task-dependent: forward KL tends to help exploration-heavy tasks (e.g., Stair) at low $\lambda$, whereas reverse KL accelerates convergence on precision-dominated tasks (e.g., Balance Simple) (Fig. 3). These results support the main claims of the work: closing the loop so that planning informs policy updates (and vice-versa) yields guided exploration and better sample efficiency in MPPI-based RL.

**Limitations.** Tuning hyperparameter $\lambda$ is essential for the performance of PO-MPC. As a rule of thumb, we keep it to $\lambda = 1$, to equally weight return maximization and KL minimization. However, its optimal value depends both on the complexity of the environment and the training of the policy prior. A similar approach might be taken as in SAC (Haarnoja et al., 2018), where the appropriate value of

*Table 2.* Method characteristics and empirical trends under the PO-MPC view. Arrows denote trends observed in our experiments; results in Table 1 and Figs. 3–5. Performance and Sample efficiency are taken w.r.t. TD-MPC2. Note that $\lambda \to \infty$ means only the KL divergence in Eq. 6 is optimized.

| Method | Uses planning policy prior | KL-reg. objective | Fwd/Rev KL (Eq. 8, 9) | Sample eff. | Final perf. |
|---|---|---|---|---|---|
| TD-MPC2 | ✗($\lambda = 0$) | ✗ | – | baseline | baseline |
| BMPC | ✓($\lambda \to \infty$) | ✓ ($\pi_P$) | Fwd | ↑ | ↑ / ≈ |
| PO-MPC (Ours) | ✓($\lambda$ var.) | ✓ ($\pi_{\theta_p}$) | Fwd / Rev | ↑ | ↑↑ |

$\lambda$ would be learned during training.

Also, information obtained during planning is not fully exploited. Many trajectories are simulated during planning that, although used for computing an action sequence, are not leveraged for learning the action value function, thus being computationally inefficient. Additionally, such trajectories are constrained to short horizons. The model loses accuracy at long horizons, which reduces the accuracy of the estimated scores for each sampled trajectory as well.

Finally, we assume both learned sampling policy and policy prior to be Gaussian distributions. This approximation is restrictive since the Planning policy, which consists of a Gaussian prior reweighted by an exponential distribution of the trajectory costs, is not necessarily Gaussian.

**Conclusion.** This paper introduced *Policy Optimization – Model Predictive Control* (PO-MPC), a family of model-based reinforcement learning methods for continuous action spaces. In particular, PO-MPC extends MPPI-based RL by finding a common formulation that includes previously published approaches in the state-of-the-art, and exploits previously unexplored design choices. Our experiments show that PO-MPC leveraging these choices often learns faster and more stably than the other baselines, serving as a new state-of-the-art for model-based RL in continuous control. Future work could focus on **1)** extending the distribution of the policies used to more expressive classes than Gaussian, **2)** automatically tuning the trade-off between Return maximization and KL minimization, and **3)** increasing efficiency by leveraging simulated data during planning.

## Acknowledgments

We would like to thank Natalia Amat and Manuel Boldrer for their valuable feedback on an early version of this paper. This work was supported by Shell.

## Impact Statement

This paper presents work whose goal is to advance the field of Machine Learning. There are many potential societal consequences of our work, none which we feel must be specifically highlighted here.

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

# A. Hyperparameters

In table 3 we share the hyperparameters employed for both our method (PO-MPC) and the baseline TD-MPC. Both methods share all parameters except for the ones exclusive to PO-MPC.

*Table 3.* Hyperparameter configuration.

| Hyperparameters | Values |
| --- | --- |
| **General** | |
| Num. steps | 1 000 000 |
| Replay buffer | 1 000 000 |
| Learning_rate | 3e-4 |
| Max. Gradient norm | 20 |
| Optimizer | Adam($\beta_1 = 0.9, \beta_2 = 0.999$) |
| **World model** | |
| Encoder dim. | 256 |
| Num. Encoder layers | 2 |
| Learning_rate | 3e-4 |
| Latent_dim | 512 |
| Dropout | 0.01 |
| Num. Value Nets | 5 |
| Num. bins | 101 |
| Symlog min,max | -10, 10 |
| Simnorm dim | 8 |
| **TD-MPC2** | |
| Horizon | 3 |
| MPPI iterations | 8 |
| Population size | 512 |
| Policy prior samples | 24 |
| Num. elites | 64 |
| Min. plan std ($\sigma_{min}$) | 0.05 |
| Max. plan std ($\sigma_{max}$) | 2 |
| Temperature | 1.0 |
| Batch size ($n_r$) | 256 |
| Discount ($\gamma$) | 0.99 |
| Time discount ($\rho$) | 0.5 |
| Consistency coef. | 20 |
| Reward model coef. | 0.1 |
| Value function coef. | 0.1 |
| Entropy coef. ($\alpha$) | 1e-4 |
| Target update coef. ($\tau$) | 0.01 |
| **PO-MPC** | |
| Biased value function coef. | 0.1 |
| KL Reg. strength $\lambda$ | {0.1, 1.0, 9.0} |
| Learned intermediate prior policy | {Yes, No} |
| Prior policy learning loss | {Fwd KL, Rev KL} |
| Reanalyzed batch ($n_b^r$) | 20 |
| Reanalyzed interval (k) | 10 |

# B. Implementation details

In this appendix, we give a thorough explanation of the procedure followed to implement PO-MPC. For the sake of completeness, we also include the explanation of MPPI for obtaining the Planning policy.

## B.1. Planning policy.

In this paper, we follow the same iterative planning process explained in Section 3 for **MPPI-based Reinforcement Learning**, where the Planning policy is iteratively refined with the help of a learned sampling policy and its associated Bootstrap action-value function. We maintain the same world model loss for the environment state encoder $h_{\theta_h}(s)$, dynamics model $p_{\theta_d}(z)$, and reward function model $r_{\theta_r}(z, a)$ over latent representations from Hansen et al. (2024).

At each time step $t$, we start planning by encoding the current state of the environment $z_t = h_\theta(s_t)$. Then we sample simulated trajectories of horizon H, sampling $n_{\pi_{s_\theta}}$ times actions from the learned sampling policy $\pi_{s_\theta}$ and $M - n_{\pi_\theta}$ times from the Planning policy. The Planning policy is a Gaussian open-loop control sequence with mean: $\bar{a}_{0:H-1} = (\bar{a}_0, \ldots, \bar{a}_{H-1})$, and every sample being computed by $a_t^{(i)} = \bar{a}_t + \epsilon_t^{(i)}$, $\epsilon_t^{(i)} \sim \mathcal{N}(0, \sigma_t I)$. The sequence is always initialized with variance $\sigma_{max}^2$, and the mean $\bar{a}_t$ with the 1-step shifted mean except for the start of the episode where zero-mean is used. $M$ noisy trajectories are simulated $z_{t+1}^{(i)} \sim p(z_{t+1} \mid z_t^{(i)}, a_t^{(i)})$ and evaluated according to its H-step estimated return:

$$\hat{Q}(z_0, a_{0:H}^{(i)}) = \sum_{t=0}^{H-1} \gamma^t r_{\theta_r}(z_t, a_t^{(i)}) + \gamma^H Q_{\theta_Q}^{\pi_{\theta_s}}(z_H, a_H^{(i)}) \tag{10}$$

After selecting the K-top performing samples, the MPPI update follows from a path-integral (desirability) transform:

$$\bar{a}_t \leftarrow \bar{a}_t + \sum_{i=1}^{K} w_i \, \epsilon_t^{(i)}, \quad \sigma_t = \sqrt{\frac{\sum_{i=1}^{K} w_i \left(\epsilon_t^{(i)}\right)^2}{\sum_{i=1}^{K} w^i}} \tag{11}$$

$$w_i = \frac{\exp\left(-\frac{1}{\beta}(Q(z_0, a_{0:H}^{(i)}) - \max_{i'} Q(z_0, a_{0:H}^{(i')}))\right)}{\sum_{j=1}^{K} \exp\left(-\frac{1}{\beta}(Q(z_0, a_{0:H}^{(j)}) - \max_{i'} Q(z_0, a_{0:H}^{(i')}))\right)},$$

where $\beta > 0$ is the temperature parameter, and controls how much the importance sampling scheme weights the optimal cost trajectory versus the others. After a fixed number of iterations, the planning procedure is terminated and a trajectory is sampled from the final return-normalized distribution over action sequences. Planning is done at each decision step, and only the first action of the sampled trajectory, $a_0$, is executed to produce a feedback policy. We denote the resulting Planning policy over the first step, obtained after a fixed number of MPPI iterations, by: $\pi_P = \mathcal{N}(\bar{a}_0, \sigma_0 I)$, with $p$ being the transition model, and $\bar{a}_{init}$ the initialization mean control sequence. After interacting with the environment, the transition information and Planning policy are added to a replay buffer, i.e. $(s, a_0, s', r, \bar{a}_0, \sigma_0) \longrightarrow \mathcal{D}$.

## B.2. Adaptive prior policy updates.

To improve the sampling policy using KL-regularized RL, we need a policy prior $\pi_p$ representing the current Planning policy to act as a reference. To represent the current Planning policy we can straightforwardly use the Planning policy samples stored in the replay buffer or, as shown in Section 4, an intermediate policy $\pi_{\theta_p}$. We train this intermediary policy by either minimizing the reverse KL divergence:

$$J(\theta_p) = \sum_{t'=t}^{H-1} \frac{\rho^{t'-t}}{H} \frac{\text{KL}[\pi_{\theta_p}(\cdot \mid z_{t'}) \parallel \pi_P(\cdot \mid z_{t'})]}{\max(1, S_p)}, \tag{12}$$

or, as an example of a straightforward alternative, the forward KL divergence:

$$J(\theta_p) = \sum_{t'=t}^{H-1} \frac{\rho^{t'-t}}{H} \frac{\text{KL}[\pi_P(\cdot \mid z_{t'}) \parallel \pi_{\theta_p}(\cdot \mid z_{t'})]}{\max(1, S_p)}, \tag{13}$$

where $S_p$ is an adaptive scale parameter that tracks the difference between the $5^{th}$ and $95^{th}$ percentiles of the KL divergence. This is often use

## B.3. Action value function and policy updates.

Planning policy improvement relies on improving the sampling policy, $\pi_{\theta_s}$, and updating its associated bootstrap action value function, $Q_{\theta_Q}^{\pi_{\theta_s}}$. Every $n_d$ time steps, a batch of $n_b$ trajectories of horizon $H$ is drawn from the replay buffer $\mathcal{D}$. The action value function $Q_{\theta_Q}^{\pi_{\theta_s}}$ is updated by minimizing its TD-error at each time step over the horizon H, with a decaying parameter $\rho$ to account for prediction error over the latent space predictions. In the following, we denote by $\pi_{\theta_s}(z)$ the learned sampling policy probability distribution over actions $u$ conditional on the latent representation $z = h_{\theta_h}(s)$, leaving $\pi_{\theta_s}(u|z)$ to denote the probability of sampling $u$ under the learned sampling policy.

$$J(\theta_Q) = \sum_{t'=t}^{H-1} \frac{\rho^{t'-t}}{H} \text{CE}(Q_{\theta_Q}^{\pi_{\theta_s}}(z_{t'}, a_{t'}), \hat{Q}^{\pi_{\theta_s}}(z_{t'}, a_{t'})) \tag{14}$$

$$\hat{Q}^{\pi_{\theta_s}}(z_{t'}, a_{t'}) = r_t + \gamma Q_{\theta_Q^-}^{\pi_{\theta_s}}(z_{t'+1}, \tilde{a})|_{\tilde{a} \sim \pi_{\theta_s}(a|z_{t'+1})} \tag{15}$$

Where $\theta_Q$ and $\theta_Q^-$ are the parameters of the action value function and the target action value function. As explained in Hansen et al. (2024), the TD-error is tracked by the cross-entropy error between action-value logit representations and the two-hot vector encoding of the target. Under the assumption that the action-value function is correctly approximated, the planning policy is a maximum a posteriori estimate over the learned sampling distribution $\pi_{\theta_s}$. Therefore, the planning policy can be intuitively interpreted as a policy improvement step over the current learned policy (Sutton & Barto, 2018).

The learned sampling policy update is designed to move the policy towards maximizing the expected return while ensuring its associated trajectory distribution remains close to the prior trajectory distribution, which is induced by the planning policy. This leads to the following KL-regularized action value function loss:

$$J(\tilde{\theta}_Q) = \sum_{t'=t}^{H-1} \frac{\rho^{t'-t}}{H} \text{CE}(Q_{\tilde{\theta}_Q}^{\pi_{\theta_s},\lambda}(z_{t'}, a_{t'}), \hat{Q}^{\pi_{\theta_s},\lambda}(z_{t'}, a_{t'})) \tag{16}$$

$$\hat{Q}^{\pi_{\theta_s},\lambda}(z_{t'}, a_{t'}) = r_t + \gamma \left( Q_{\tilde{\theta}_Q^-}^{\pi_{\theta_s},\lambda}(z_{t'+1}, \tilde{a})|_{\tilde{a} \sim \pi_{\theta_s}(z_{t'+1})} - \lambda \frac{\text{KL}[\pi_{\theta_s}(\cdot|z_{t'+1}) \| \pi_{\theta_p}(\cdot|z_{t'+1})]}{\max(1, S_{KL})} \right) \tag{17}$$

and the following policy loss:

$$J(\theta_s) = \sum_{t'=t}^{H-1} \frac{\rho^{t'-t}}{H} \left( \lambda \frac{\text{KL}[\pi_{\theta_s}(z_{t'}) \| \pi_{\theta_p}(z_{t'})]}{\max(1, S_{KL})} - \frac{Q_{\tilde{\theta}_Q}^{\pi_{\theta_s},\lambda}(z_{t'}, \tilde{a})|_{\tilde{a} \sim \pi_{\phi_\pi}(z_{t'})}}{\max(1, S_Q)} - \alpha \mathcal{H}(\pi_{\theta_s}(z_t)) \right), \tag{18}$$

where $S_i$, $i \in \{KL, Q\}$, is an adaptive scale parameter that tracks the difference between the $5^{th}$ and $95^{th}$ percentiles of each loss term. Since the values of both terms differ by multiple degrees of magnitude, scaling them enables more robust control, through the hyperparameter $\lambda$, over the trade-off between expected return maximization and mimicking the policy prior distribution.

It is important to note that, due to its potential to reach very high values, which may negatively affect action value learning and, consequently, exploration, the KL term, both in action value target and sampling policy update, is often scaled by $S_{KL}$ in practice.

## B.4. Co-dependence between the learned policy and the planning policy.

During the first steps of training, the replay buffer needs to be filled, and the planning policy suffers from low quality since both $Q_{\theta_Q}^{\pi_{\theta_s}}, \pi_{\theta_s}$ are untrained. This is why it is important to make sure the bootstrap action value function is properly trained

before updating all the other components. Therefore, we follow a pretraining phase during the first $N_s$ steps, where only the untrained sampling policy $\pi_{\theta_s}$ interacts with the environment with no parameter updates. Then, before proceeding to update all parameters as explained in Section 4, we update all model parameters and the bootstrapping action value function ($Q_{\theta_Q}^{\pi_{\theta_s}}$) $N_s$ times. To prevent unnecessary exploration bias, the planning policy samples stored during this phase are zero-mean diagonal Gaussians with maximum standard deviation $\sigma_{max}$. This ties with another relevant implementation detail. Due to the planning policy depending on an ever-evolving policy distribution, planning policy samples saved in the replay buffer eventually become outdated. To alleviate this problem, we employ *lazy reanalyze* (Wang et al., 2025b), which takes inspiration from the (Wang et al., 2024; Schrittwieser et al., 2021) to periodically update partially a subset of the planning distributions sampled from the replay buffer.

### B.5. Architecture and Framework

In this work, we build upon the partial implementation of TD-MPC2 in JAX (Bradbury et al., 2018) by Flandermeyer (2024a). We inherit all architectural choices from TD-MPC2. The architecture of $Q_{\hat{\theta}_Q}^{\pi_{\theta_s}, \lambda}$ follows the same design of its counterpart $Q_{\theta_Q}^{\pi_{\theta_s}}$. Despite updating an additional policy and action value function, training times do not differ significantly from the baselines.

### B.6. Baselines.

For our experiments, we employ the implementations in JAX (Flandermeyer, 2024a;b), developed with the collaboration of the original authors, since they reproduce the results from the original paper while increasing the computation speed.

# C. PO-MPC algorithm

**Algorithm 2** PO-MPC

**Require:** Replay buffer $\mathcal{B}$, Data-to-update ratio $n_{d2u}$, and Reanalyze interval $k$.

    **Initialize**: $\pi_{\theta_s}, Q^{\pi}_{\theta_Q}, Q^{\pi_{\theta_s},\lambda}_{\tilde{\theta}_Q}$.

    **Initialize** MDP model: $\tilde{\mathcal{M}} := (h_{\theta_h}, p_{\theta_d}, r_{\theta_r})$.

    **Initialize** planning priors: $a_{init}, \sigma_{max}$

1:  n_updates $= 0$
2:  **for** t=1,2,...,T **do**
3:    *// Environment interaction*
4:    $z_t \leftarrow h_{\theta_h}(s_t)$
5:    *// Planning Policy (Section B.1)*
6:    $a_t, \bar{a}_{t:t+H}, \sigma_{t:t+H} \leftarrow \text{MPPI}_{\tilde{\mathcal{M}}}(z_t | \pi_{\theta_s}, Q^{\pi_{\theta_s}}_{\theta_Q}, \bar{a}_{init})$
7:    $s_{t+1}, r_t \leftarrow \text{environment\_step}(s_t, a_t)$
8:    $\mathcal{B} \cup \{s_t, a_t, r_t, s_{t+1}, \bar{a}_t, \sigma_t\}$
9:    *// Gradient updates.*
10:    **if** t (mod $n_{d2u}$) $== 0$ **then**
11:      n_updates $\leftarrow$ n_updates $+ 1$
12:      $\mathcal{D}_{n_b} := \{s_{t'}, a_{t'}, r_{t'}, s_{t'+1}, \bar{a}_{t'}, \sigma_{t'}\}^{1:n_b}_{t':t'+H} \sim \mathcal{D}$
13:      $z_{t':t'+H} \leftarrow h_{\theta_h}(s_{t':t'+H})$
14:      *// Update Planning samples via Lazy reanalyze as in Wang et al. (2025b).*
15:      **if** n_updates (mod $k$) $== 0$ **then**
16:        $\mathcal{D}_{n^r_b} \sim \mathcal{D}_{n_b}, n^r_b \leq n_b$
17:        $a_{t'}, \bar{a}_{t':t'+H}, \sigma_{t':t'+H} \leftarrow \text{MPPI}_{\tilde{\mathcal{M}}}(z_{t'} | \pi_{\theta_s}, Q^{\pi_{\theta_s}}_{\theta_Q}, \mathbf{0})$
18:        $\mathcal{D}_{n^r_b} \leftarrow \{s_{t'}, u_{t'}, r_{t'}, a_{t'+1}, \bar{a}_{t'}, \sigma_{t'}\}$
19:      **end if**
20:      $\pi_P(a_{t'} | z_{t'}) \leftarrow \mathcal{N}(\bar{a}_{t'}, \sigma^2_{t'} I)$
21:      Update MDP model: $h_{\theta_h}, d_{\theta_d}, r_{\theta_r}$ as in Hansen et al. (2024).
22:      Update Bootstrap action value function: $Q^{\pi_{\theta_s}}_{\theta_Q}$ (Equation 14)
23:      Update Policy prior: $\pi_{\theta_p}$ (Equation 8 or Equation 9
24:      Update KL regularized action value function: $Q^{\pi_{\theta_s},\lambda}_{\tilde{\theta}_Q}$ (Equation 16)
25:      Update Sampling Policy $\pi_{\theta_s}$ (Equation 18)
26:      $\theta^-_Q \leftarrow \tau\theta_Q + (1-\tau)\theta^-_Q$
27:      $\tilde{\theta}^-_Q \leftarrow \tau\tilde{\theta}_Q + (1-\tau)\tilde{\theta}^-_Q$
28:    **end if**
29:  **end for**

# D. Additional Results

## D.1. Learning curves in DMControl Suite

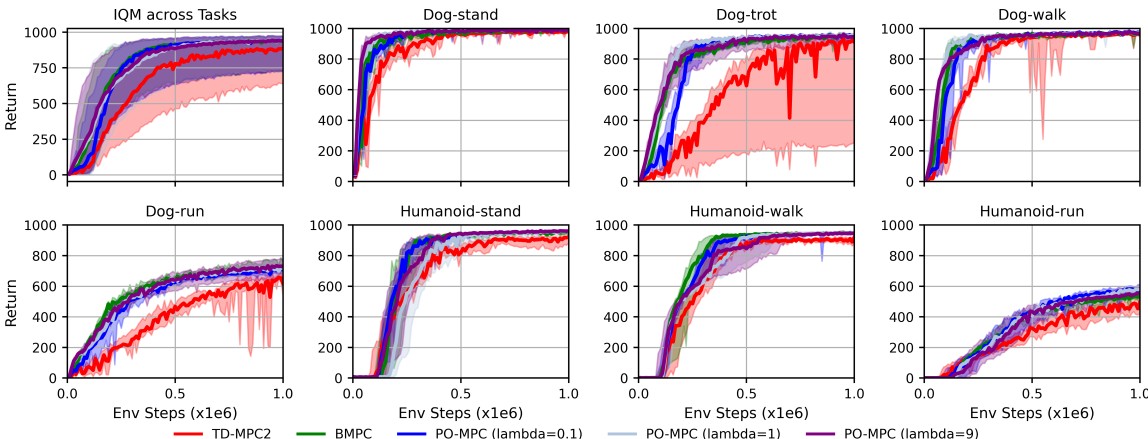

*Figure 4.* Performance comparison of PO-MPC and the baselines on 7 state-based high-dimensional control tasks from DMControl Suite (Tassa et al., 2018). Interquartile Mean (IQM) of 5 runs; shaded areas are 95% bootstrap Confidence Intervals (CI). In the top left, we visualize the Aggregate IQM with stratified bootstrap CI across all 7 tasks.

## D.2. Learning curves in HumanoidBench

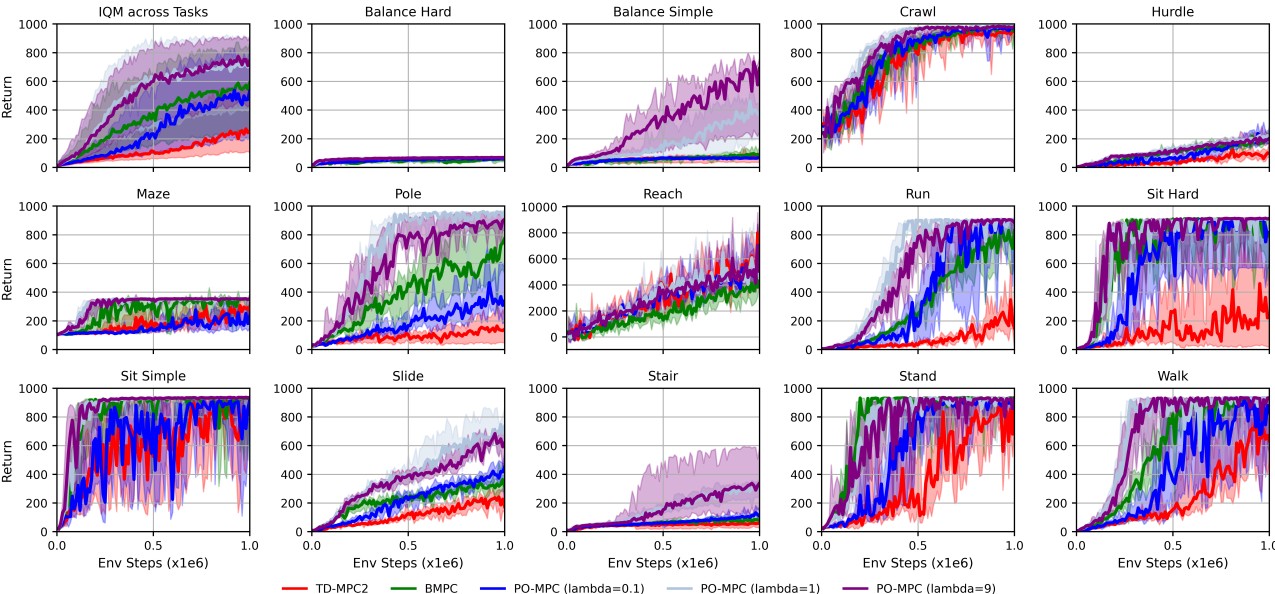

*Figure 5.* Performance comparison in 14 state-based high-dimensional control tasks from HumanoidBench (Sferrazza et al., 2024). Interquartile Mean (IQM) of 5 runs; shaded areas are the 95% bootstrap Confidence Intervals (CI). In the top left, we visualize the Aggregate IQM with stratified bootstrap CI across all tasks except for *Reach*, which has a different return range.

## D.3. Intermediate Policy Prior Performance

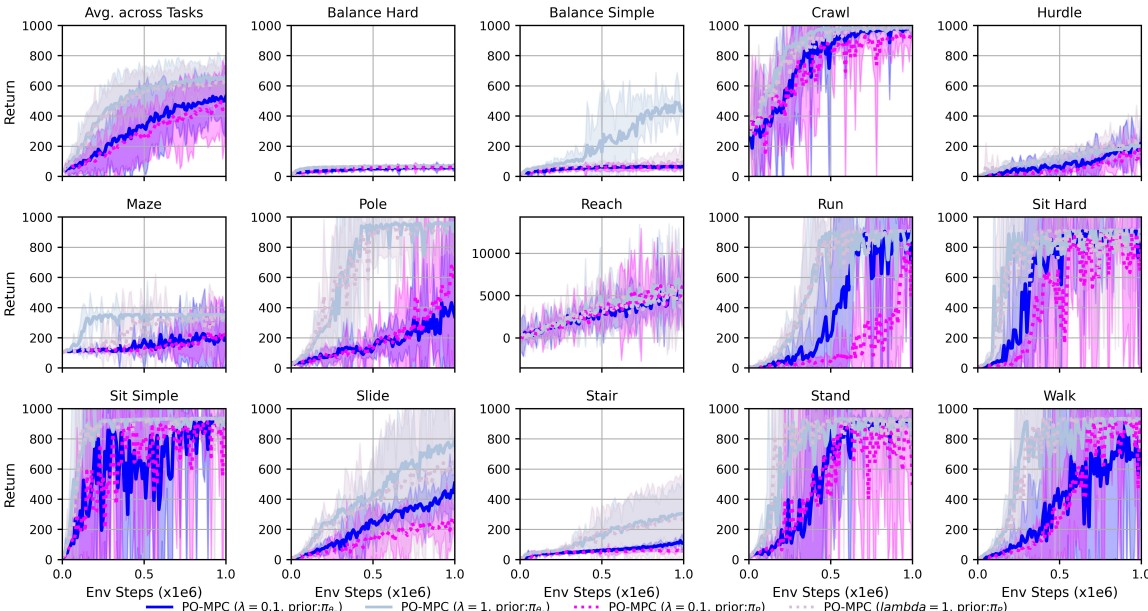

*Figure 6.* Performance comparison in 14 state-based high-dimensional control tasks from HumanoidBench (Sferrazza et al., 2024). Mean of 3 runs; shaded areas are the 95% confidence intervals. In the top left, we visualize results averaged across all tasks except for *Reach*, which has a different return range. We observe that using the intermediate policy not only does not harm the performance but also enhances it in some tasks.

### D.3.1. Shielding Effect of the Policy Prior

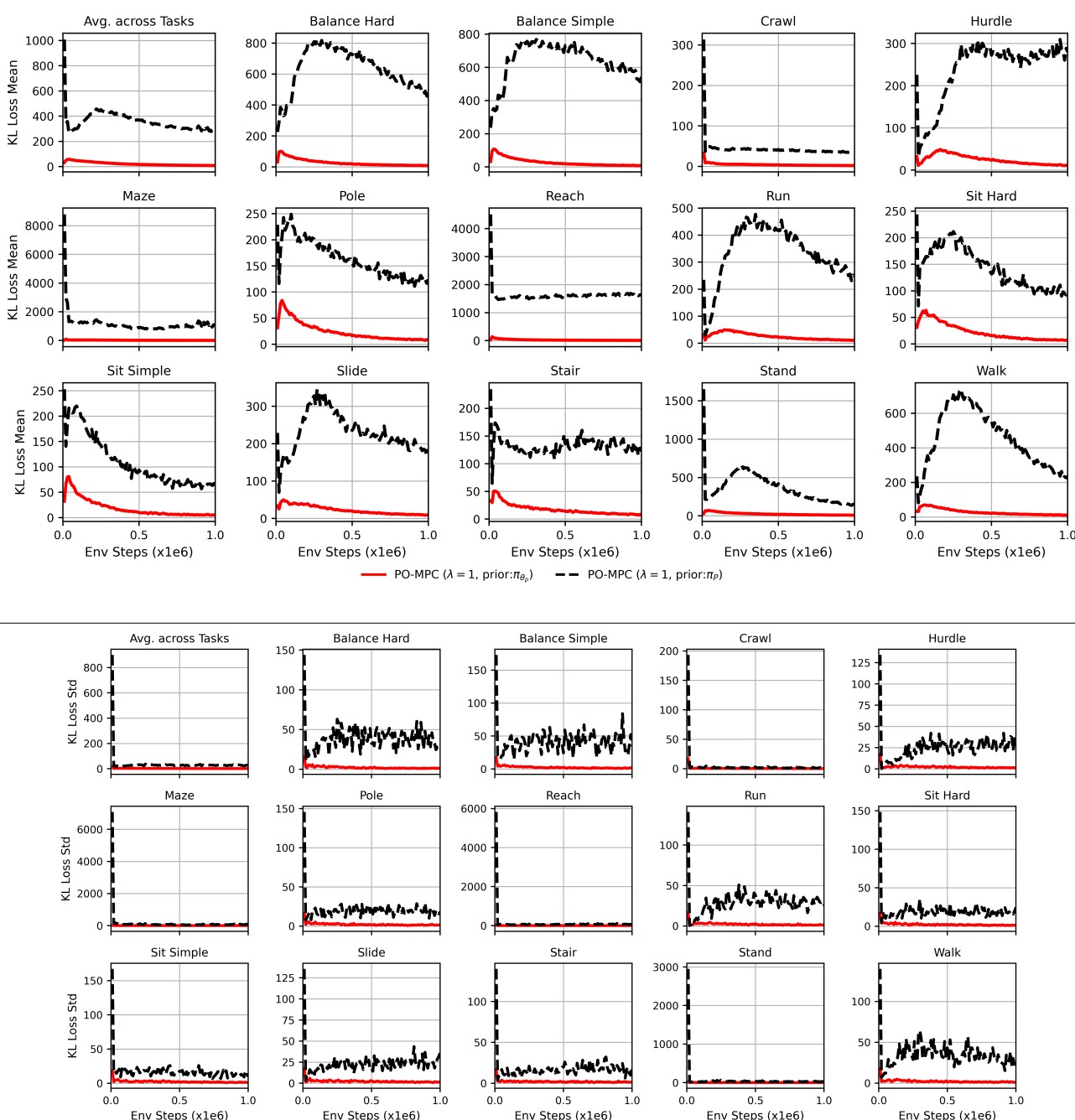

*Figure 7.* **Top:** Mean and, **Bottom:** Standard deviation of the KL divergence term in Equation 18 for both PO-MPC using an intermediate policy prior and the Planning policy. Experiments are done in the HumanoidBench Locomotion suite (Sferrazza et al., 2024). Mean of 3 runs. We show empirical evidence on how the mean and standard deviation of the KL term are significantly larger when the Planning policy samples are used instead of the intermediate policy prior. This shows that the intermediate policy prior effectively shields the sampling policy updates from high variance being introduced by outdated Planning policy samples stored in the replay buffer. Similar results are obtained across different values of $\lambda$, and we present results for $\lambda = 1$ for the sake of clarity.

### D.3.2. TRAINING POLICY PRIOR WITH REVERSE KL VS FORWARD KL LOSS

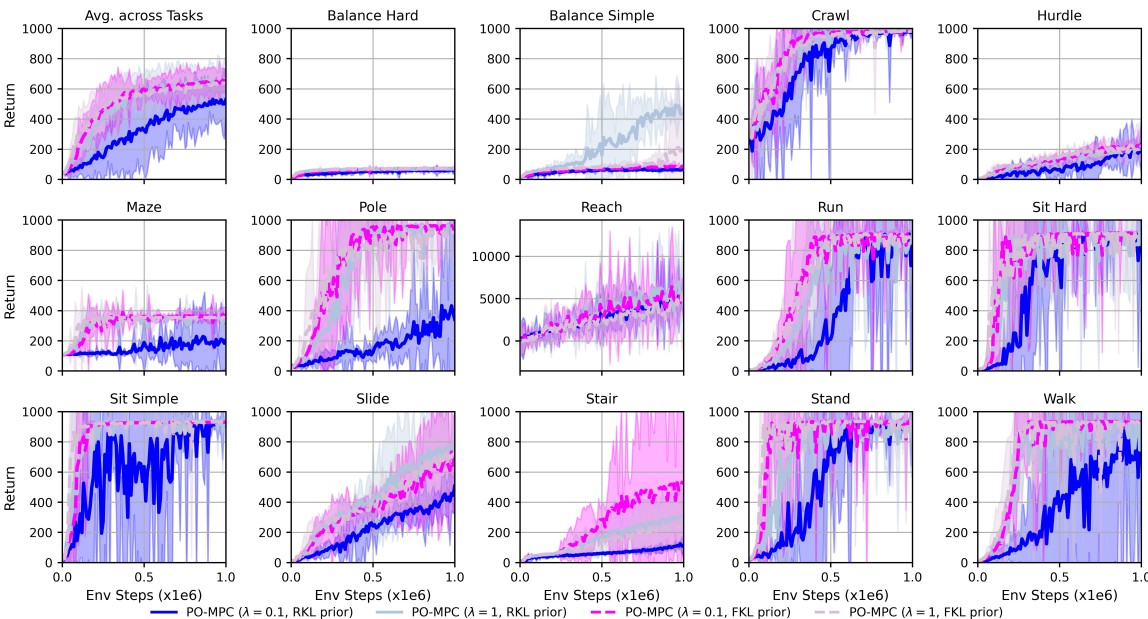

*Figure 8.* Performance comparison in 14 state-based high-dimensional control tasks from HumanoidBench Locomotion suite (Sferrazza et al., 2024). Mean of 3 runs; shaded areas are 95% confidence intervals. In the top left, we visualize results averaged across all tasks except for *Reach*, which has a different return range. We observe that training the policy prior with the Forward KL divergence instead of the Reverse KL divergence can help in finding a solution faster in some tasks but may be detrimental in others requiring more precision such as *Balance Simple*.

# E. Proofs

We present proof of convergence of both the KL-regularized Policy Evaluation step and policy improvement. We follow closely the same proofs for Max. Entropy RL from (Haarnoja et al., 2018) since it is a particular case of KL-regularized RL. Substituting $\pi_p$ by the uniform distribution over the action space $\mathcal{A}$ recovers the proof from (Haarnoja et al., 2018). Note that the only additional requirement needed for the KL-regularized version is $\frac{\pi(a_t|s_t)}{\pi_p(a_t|s_t)}$ being determined almost everywhere.

**Lemma E.1.** *(KL-regularized Policy Evaluation). Given a policy and policy prior $\pi, \pi_p \in \Pi$. Let the KL-regularized Bellman backup operator:*

$$\mathcal{T}^\pi Q^{\pi,\lambda}(s_t, a_t) := r(s_t, a_t) + \gamma \mathbb{E}_{s_{t+1} \sim p(\cdot|s_t, a_t)}[V^{\pi,\lambda}(s_{t+1})] \tag{19}$$

*where*

$$V^{\pi,\lambda}(s_t) = \mathbb{E}_{a_t \sim \pi(\cdot|s_t)}[Q^{\pi,\lambda}(s_t, a_t) - \lambda[\log \pi(a_t|s_t) - \log \pi_p(a_t|s_t)]], \tag{20}$$

*and a mapping $Q_0 : \mathcal{S} \times \mathcal{A} \longrightarrow \mathbb{R}$, where $\mathcal{A}$ is bounded and $\frac{\pi(a_t|s_t)}{\pi_p(a_t|s_t)}$ is determined almost everywhere, we define $Q_{k+1}^{\pi,\lambda} = \mathcal{T}^\pi Q_k^{\pi,\lambda}$. Then the sequence $Q_k^{\pi,\lambda}$ will converge to the KL regularized action-value of $\pi$ as $k \to \infty$.*

*Proof.* Define the KL augmented reward as:

$$r_\pi := r(s_t, a_t) - \gamma\lambda \mathbb{E}_{s_{t+1} \sim p(\cdot|s_t, a_t)}[\text{KL}[\pi(\cdot \mid s_{t+1}) \parallel \pi_p(\cdot \mid s_{t+1})]] \tag{21}$$

and rewrite the update as:

$$Q^{\pi,\lambda}(s_t, a_t) \leftarrow r_\pi(s_t, a_t) + \gamma \mathbb{E}_{\substack{s_{t+1} \sim p(\cdot|s_t, a_t), \\ a \sim \pi(\cdot|s_{t+1})}}[Q^{\pi,\lambda}(s_{t+1}, a_{t+1})] \tag{22}$$

Then we can apply the standard convergence results for policy evaluation from (Sutton & Barto, 2018). A bounded action space $\mathcal{A}$ and KL-divergence between $\pi$ and $\pi_p$ are necessary assumptions to guarantee that the augmented reward $r_\pi$ is bounded. $\square$

**Lemma E.2.** *(KL-regularized Policy Improvement) Let $\pi_{old} \in \Pi$ and let $\pi_{new}$ be the optimizer of the minimization problem defined as:*

$$\pi_{new} = \arg \min_{\pi' \in \Pi} KL[\pi'(\cdot|s_t) \parallel \frac{\exp(\frac{1}{\lambda} Q^{\pi_{old},\lambda}(s_t, \cdot))}{Z^{\pi_{old}}(s_t)} \pi_p(\cdot|s_t)] = \arg \min_{\pi' \in \Pi} \mathcal{J}_{\pi_{old}}(\pi'(\cdot|s_t)) \tag{23}$$

*Then $Q^{\pi_{new},\lambda}(s_t, a_t) \geq Q^{\pi_{old},\lambda}(s_t, a_t), \forall(s_t, a_t) \in \mathcal{S} \times \mathcal{A}$ with $|\mathcal{A}| < \infty$ being bounded.*

*Proof.* Let $\pi^{old} \in \Pi$, and $Q^{\pi_{old},\lambda}, V^{\pi_{old},\lambda}$, its respective KL-regularized action-value and value function. Then we define:

$$\pi_{new} = \arg \min_{\pi' \in \Pi} \mathcal{J}_{\pi_{old}}(\pi'(\cdot|s_t))$$
$$= \arg \min_{\pi' \in \Pi} \text{KL}[\pi'(\cdot|s_t) \parallel \exp(\lambda^{-1} Q^{\pi_{old},\lambda}(s_t, \cdot) - \log Z^{\pi_{old}}(s_t) + \log \pi_p(\cdot|s_t))] \tag{24}$$

It must be the case that $\mathcal{J}_{\pi_{old}}(\pi_{new}(\cdot|s_t)) \leq \mathcal{J}_{\pi_{old}}(\pi_{old}(\cdot|s_t))$, since we can always choose $\pi_{new} = \pi_{old} \in \Pi$. Hence,

$$\mathbb{E}_{a_t \sim \pi^{new}}\left[\log \pi^{new}(a_t|s_t) - \lambda^{-1} Q^{\pi_{old},\lambda}(s_t, a_t) + \log Z^{\pi_{old}}(s_t) - \log \pi_p(a_t|s_t)\right] \leq$$
$$\mathbb{E}_{a_t \sim \pi^{old}}\left[\log \pi^{old}(a_t|s_t) - \lambda^{-1} Q^{\pi_{old},\lambda}(s_t, a_t) + \log Z^{\pi_{old}}(s_t) - \log \pi_p(a_t|s_t)\right]. \tag{25}$$

Since $Z^{\pi^{old}}$ does not depend on $a_t$, equation 25 reduces to:

$$\mathbb{E}_{a_t \sim \pi^{new}}\left[Q^{\pi_{old},\lambda}(s_t, a_t) - \lambda \log \frac{\pi^{new}(a_t|s_t)}{\pi_p(a_t|s_t)}\right] \geq$$
$$\mathbb{E}_{a_t \sim \pi^{old}}\left[Q^{\pi_{old},\lambda}(s_t, a_t) - \lambda \log \frac{\pi^{old}(a_t|s_t)}{\pi_p(a_t|s_t)}\right] = V^{\pi_{old},\lambda}(s_t). \tag{26}$$

Then, unrolling $Q^{\pi_{old},\lambda}(s_t, a_t)$ and applying the bound in equation 26 results in:

$$
\begin{aligned}
Q^{\pi_{old},\lambda}(s_t, a_t) &= r(s_t, a_t) + \gamma \mathbb{E}_{s_{t+1} \sim p(\cdot|s_t,a_t)}[V^{\pi_{old},\lambda}(s_{t+1})] \\
&\leq r(s_t, a_t) + \gamma \mathbb{E}_{\substack{s_{t+1} \sim p(\cdot|s_t,a_t) \\ a_{t+1} \sim \pi_{new}(\cdot|s_{t+1})}} \left[ Q^{\pi_{old},\lambda}(s_{t+1}, a_{t+1}) - \lambda \log \frac{\pi^{new}(a_{t+1}|s_{t+1})}{\pi_p(a_{t+1}|s_{t+1})} \right] \\
&= r(s_t, a_t) + \gamma \mathbb{E}_{\substack{s_{t+1} \sim p(\cdot|s_t,a_t) \\ a_{t+1} \sim \pi_{new}(\cdot|s_{t+1})}} \left[ r(s_{t+1}, a_{t+1}) - \lambda \log \frac{\pi^{new}(a_{t+1}|s_{t+1})}{\pi_p(a_{t+1}|s_{t+1})} \right. \\
&\qquad\qquad\qquad\qquad\qquad\qquad \left. + \gamma \mathbb{E}_{s_{t+2} \sim p(\cdot|s_{t+1},a_{t+1})}[V^{\pi_{old},\lambda}(s_{t+2})] \right] \\
&\vdots \\
&\leq Q^{\pi_{new},\lambda}(s_t, a_t)
\end{aligned}
\tag{27}
$$

Convergence to $Q^{\pi_{new},\lambda}$ follows from Lemma E.1

$\square$

**Theorem E.3.** *(KL-regularized policy iteration). Repeated application of KL-regularized policy evaluation and KL-regularized policy improvement to any $\pi \in \Pi$ converges to a policy $\pi^*$ such that $Q^{\pi^*,\lambda}(s_t, a_t) \geq Q^{\pi,\lambda}(s_t, a_t)$ for all $\pi \in \Pi$ and $(s_t, a_t) \in \mathcal{S} \times \mathcal{A}$, assuming $|\mathcal{A}| < \infty$.*

*Proof.* The proof follows the same reasoning from Theorem 1 in Haarnoja et al. (2018). Let policy $\pi_i$ be the policy at iteration $i$. By Lemma E.2, the sequence $Q^{\pi_i,\lambda}$ is monotonically increasing. Since $Q^{\pi,\lambda}$ is bounded above for $\pi \in \Pi$, since both reward and KL-divergence are bounded, the sequence converges to some $\pi^*$. To show that $\pi^*$ is optimal, it must be the case that, at convergence, $J_{\pi^*}(\pi^*(\cdot|s_t)) < J_{\pi^*}(\pi(\cdot|s_t)), \forall \pi \in \Pi, \pi \neq \pi^*$. Using the same iterative argument as in the proof of Lemma E.2, we get $Q^{\pi^*,\lambda}(s_t, a_t) > Q^{\pi,\lambda}(s_t, a_t), \forall (s_t, a_t) \in \mathcal{S} \times \mathcal{A}$, meaning the KL-regularized value of any other policy in $\Pi$ is lower than that of the converged policy. Hence $\pi^*$ is optimal in $\Pi$. $\square$

# F. Background and Connection to Recent Work

There is some recent work that, while also within MPPI-based RL, cannot be unified within the proposed framework as seamlessly since they constitute an approximation of our theoretical framework(Lin et al., 2025), forgoing the monotonic improvement guarantees shown in Appendix E, or optimize a different cost function (Zhuang et al., 2025). We will start this section by briefly reviewing the background of RL cast as a probabilistic inference problem, which ultimately boils down to the KL-regularized RL formulation, since then the main differences with respect to these works will become clearer.

## F.1. RL cast as probabilistic inference

We follow the same reasoning as in (Levine, 2018). Let $\tau := (s_0, a_0, s_1, a_1, ..., s_T)$ a trajectory across the joint state-action space $\mathcal{S} \times \mathcal{A}$, the probability of a trajectory $\tau$ given a parametric policy $\pi_\theta$:

$$p(\tau; \pi_\theta) = \rho_0(s_0) \prod_{t=0}^{T} p(s_{t+1}|s_t, a_t) \pi_\theta(a_t|s_t) \tag{28}$$

Next, we assume that the probability of $(s, a)$ being part of an optimal trajectory is proportional to $\exp(\frac{1}{\lambda} r(s_t, a_t))$. Then, it follows that the joint probability of a trajectory given a prior policy $\pi_p$ and that trajectory being optimal is given by:

$$p(\tau, O_{0:T}; \pi_p) = \rho_0(s_0) \frac{1}{Z_t} \exp(\frac{1}{\lambda} R_t) \prod_{t=0}^{T} p(s_{t+1}|s_t, a_t) \pi_p(a_t|s_t) \tag{29}$$

Where $\rho_0(s_0)$ is the probability of starting at state $s_0$, $p(s_{t+1}|s_t, a_t)$ is the transition probability, and $\lambda$ is the inverse temperature. Note that $\lambda$ trades off the effect of the policy prior and the trajectory's return in the joint probability distribution. The event of $\tau$ being an optimal trajectory is represented by $O_{0:T}$ (Levine, 2018), and $R_t = \sum_{t=0}^{T} r(s_t, a_t)$. Then, the KL-regularized RL formulation presented in Equation 5 stems from minimizing **the reverse KL-divergence between** $p(\tau; \pi_\theta)$ **and** $p(\tau, O_{0:T}; \pi_p)$:

$$\theta^* = \arg \min_\theta \text{KL}[p(\tau; \pi_\theta) \parallel p(\tau, O_{0:T}; \pi_p)] \tag{30}$$

$$= \arg \min_\theta \mathbb{E}_{\substack{s_0 \sim \rho_0, a_t \sim \pi_\theta(\cdot|s_t), \\ s_{t+1} \sim p(\cdot|s_t, a_t)}} \left[ \log \frac{p(\tau; \pi_\theta)}{p(\tau, O_{0:T}; \pi_p)} \right] \tag{31}$$

$$= \arg \max_\theta \mathbb{E}_{\substack{s_0 \sim \rho_0, a_t \sim \pi_\theta(\cdot|s_t), \\ s_{t+1} \sim p(\cdot|s_t, a_t)}} \left[ \sum_{t=0}^{T-1} \left[ r(s_t, a_t) - \lambda \log \frac{\pi_\theta(a_t \mid s_t)}{\pi_p(a_t \mid s_t)} \right] \right] \tag{32}$$

$$= \arg \max_\theta \mathbb{E}_{\substack{s_0 \sim \rho_0, a_t \sim \pi_\theta(\cdot|s_t), \\ s_{t+1} \sim p(\cdot|s_t, a_t)}} \left[ \sum_{t=0}^{T-1} \left[ r(s_t, a_t) - \lambda \text{KL}[\pi_\theta(\cdot \mid s_t) \parallel \pi_p(\cdot \mid s_t)] \right] \right], \tag{33}$$

Which results in the step-wise objective described in Equation 6 with the KL-regularized action-value function $Q^{\pi_\theta, \lambda}$ defined in Equation 7.

## F.2. TD-M(PC)$^2$ (Lin et al., 2025)

The previous policy update is core to the PO-MPC framework and is guaranteed to monotonically improve under the assumptions given in Appendix E. It also includes existing methods in the literature (i.e. TD-MPC2 (Hansen et al., 2024), BMPC (Wang et al., 2025b)) when learning the sampling policy $\pi_{\theta_s}$ under different values of the hyperparameter $\lambda$ and a policy prior that represents the planner policy, either from data stored in the replay buffer $\pi_p = \pi_P$ or a proxy distribution $\pi_p = \pi_{\theta_p}$.

However, it adds further complexity since it requires learning an additional action value function $Q^{\pi_\theta, \lambda}$. Therefore, **recent methods such as TD-M(PC)$^2$**, choose to forgo theoretical guarantees in favor of simplicity by using the unregularized $Q^\pi$, which is still bound to perform well as long as the policy remains close to the planner. This is enforced by maximizing:

$$\mathbb{E}_{a \sim \pi_{\theta_s}}[Q^{\pi_{\theta_s}}(z_t, a_t) - \alpha \log \pi(\cdot \mid z_t) + \beta \log \pi_p(\cdot \mid z_t)] \tag{34}$$

Where $\alpha$ is the hyperparameter regulating entropy maximization and $\beta$ modulates the cross-entropy These hyperparameters are often tuned so that $\alpha \ll \beta$, which allows to further develop this expression into:

$$\mathbb{E}_{a_t \sim \pi_{\theta_s}}\Big[Q^{\pi_{\theta_s}}(z_t, a_t) - \alpha \log \pi_{\theta_s}(a_t \mid z_t) + \beta \log \pi_p(a_t \mid z_t)\Big]$$

$$= \mathbb{E}_{a_t \sim \pi_{\theta_s}}\Big[Q^{\pi_{\theta_s}}(z_t, a_t) + (\beta - \alpha) \log \pi_{\theta_s}(a_t \mid z_t) - \beta \log \frac{\pi_{\theta_s}(a_t \mid z_t)}{\pi_p(a_t \mid z_t)}\Big] \tag{35}$$

Which can be seen as learning a policy that maximizes the action value function $Q^{\pi_{\theta_s}}$ while minimizing both KL-divergence with respect to $\pi_p$, and entropy, since $\beta - \alpha$ is positive under $\alpha \ll \beta$.

### F.3. TD-MPBC (Zhuang et al., 2025) and BOOM (Zhan et al., 2025)

Other recent works in MPPI-based RL cannot be included within our formulation because the starting loss function being minimized is different. The policy updates from TD-MPBC (Zhuang et al., 2025) and BOOM (Zhan et al., 2025) comes from minimizing **the forward KL-divergence between** $p(\tau; \pi_\theta)$ **and** $p(\tau, O_{0:T}; \pi_p)$:

$$\theta^* = \arg \min_\theta \mathrm{KL}[p(\tau, O_{0:T}; \pi_p) \parallel p(\tau; \pi_\theta)]$$

$$= \arg \min_\theta \mathbb{E}_{(s_t, a_t) \sim p(\tau, O_{0:T}; \pi_p)}\Big[\log \frac{p(\tau, O_{0:T}; \pi_p)}{p(\tau; \pi_\theta)}\Big]$$

$$= \arg \max_\theta \mathbb{E}_{(s_t, a_t) \sim p(\tau, O_{0:T}; \pi_p)}\Big[\log p(\tau; \pi_\theta)\Big]$$

$$= \arg \max_\theta \mathbb{E}_{(s_t, a_t) \sim p(\tau, O_{0:T}; \pi_p)}\Big[\sum_{t=0}^{T-1} \log \pi_\theta(a_t \mid s_t)\Big]$$

$$= \arg \max_\theta \mathbb{E}_{(s_t, a_t) \sim p(\tau; \pi_p)}\Big[\sum_{t=0}^{T-1} \frac{1}{Z_t} \exp(\frac{1}{\lambda} R_t) \log \pi_\theta(a_t \mid s_t)\Big]$$

$$= \arg \max_\theta \mathbb{E}_{(s_t, a_t) \sim p(\tau; \pi_p)}\Big[\sum_{t=0}^{T-1} \exp(\frac{1}{\lambda} R_t - \log Z_t) \log \pi_\theta(a_t \mid s_t)\Big] \tag{36}$$

Since in this case $\pi_p$ can be an arbitrary policy, we can estimate $\mathbb{E}_{(s_t, a_t) \sim p(\tau; \pi_p)}[\cdot]$ through Monte Carlo estimation. Choosing $\lambda = G$ and $\log Z_t = 1$, we recover the Behavior Cloning regularization term from TD-MPBC:

$$\arg \max_\theta \sum_{(s_t, a_t, R_t) \sim \mathcal{D}} \exp\Big(\frac{R_t - G}{G}\Big) \log \pi_\theta(a_t \mid s_t) \tag{37}$$

This term has high variance, since it depends on the returns $R_t$ from the full trajectory. It is also overly reliant on obtaining high-return trajectory samples, which might be difficult at the start of training. Choosing instead to approximate $R_t$ with an action value function $Q(s_t, a_t)$, leave $\lambda$ as a hyperparameter, and replace $Z_t = \sum_{(s_t, a_t) \sim \mathcal{D}} \exp\left(\frac{1}{\lambda} Q(s_t, a_t)\right)$ results in:

$$\arg \max_\theta \sum_{(s_t, a_t) \sim \mathcal{D}} \frac{\exp\left(\frac{1}{\lambda} Q(s_t, a_t)\right)}{\sum_{(s_t, a_t) \sim \mathcal{D}} \exp\left(\frac{1}{\lambda} Q(s_t, a_t)\right)} \log \pi_\theta(a_t \mid s_t) \tag{38}$$

,

which corresponds to the soft Q-weighted alignment loss from BOOM (Zhan et al., 2025).

In both losses, the training signal is potentially sparse, as they both assign a high weight to the samples from the replay buffer with high expected return and ignore the rest. This is possibly why the main learning signal in TD-MPBC and BOOM

originates from using the policy updates from TD-MPC2 (equivalent to PO-MPC with $\lambda = 0$), with the term in Equation 37 and 38 serving as a regularization term.

### F.3.1. COMPARISON TO BOOM (ZHAN ET AL., 2025)

*Table 4.* Comparison of the final average return of BOOM (Zhan et al., 2025) and PO-MPC, for intermediate $\lambda$ values, after training for 1M steps. Evaluations across 7 high-dimensional control tasks from DMControl Suite (Tassa et al., 2018), and 14 from Humanoid-Bench (Sferrazza et al., 2024). Results of PO-MPC are extracted from the same data used to generate Table 1, and the results in Figures 4 and 5: Mean of 5 runs and 95% CI. The results of BOOM are publicly available on the method's GitHub repository: Mean of 3 runs and 95% CI. **Bold** is best. Higher is better.

| Task | BOOM | Ours ($\lambda$=0.1) | Ours ($\lambda$=1) | Ours ($\lambda$=9) |
|---|---|---|---|---|
| Dog-stand | 925±51 | 990 ± 4 | **992 ± 2** | 988 ± 3 |
| Dog-trot | 863±46 | **951 ± 17** | 944 ± 5 | 940 ± 29 |
| Dog-walk | 952±4 | **974 ± 4** | 968 ± 5 | 967 ± 12 |
| Dog-run | 653±36 | 708 ± 20 | **721 ± 38** | 716 ± 54 |
| Humanoid-stand | 911±17 | 957 ± 4 | 954 ± 9 | **960 ± 4** |
| Humanoid-walk | 901±20 | 946 ± 5 | **948 ± 2** | 947 ± 6 |
| Humanoid-run | 443±18 | **580 ± 27** | 568 ± 36 | 567 ± 30 |
| H1hand-hurdle | 185±38 | **226 ± 55** | 219 ± 36 | 204 ± 29 |
| H1hand-pole | 688±88 | 355 ± 198 | **940 ± 48** | 900 ± 64 |
| H1hand-run | 330±11 | 900 ± 6 | **907 ± 3** | 857 ± 79 |
| H1hand-sit-s. | 897±10 | 919 ± 8 | 933 ± 3 | **934 ± 5** |
| H1hand-slide | 471±63 | 394 ± 200 | **742 ± 116** | 595 ± 96 |
| H1hand-stand | 909±32 | 881 ± 43 | 926 ± 14 | **928 ± 14** |
| H1hand-walk | 900±31 | 809 ± 141 | **910 ± 49** | 830 ± 233 |

# G. Theoretical Justification over the Adaptive Prior

This section yields a theoretical justification for why using a learned policy prior that minimizes the KL divergence with the previously stored planner policy samples from the replay buffer, instead of using these directly, may decrease gradient variance from the sampling policy updates. First, we introduce the formal definition for the gradient related to the regularization term that results from using each representation of the planner policy. Then we compute the variance of each and compare them.

Let us sample transitions from 1 to $K$, $K$ being the current time step, and let $\mathcal{K} := \{1, 2, ..., K\}$ the set of previous time steps. Let $(s_k, a_k, s_{k+1}, r_k, \mu_k, \sigma_k)$ be the transition stored at time step $k$, where $\mu_k, \sigma_k$ are the mean and standard deviation of the planner policy computed at time step $k$: $\pi^k := \mathcal{N}(\mu_k, \sigma_k^2 I)$.

Assume that we can isolate $N << K$ samples that share the same state $s$ and compute the mean KL-divergence between the parametric sampling policy we want to update, $\pi_\theta$ (where $\theta = \theta_s$ for simplicity), and some generic prior policy, $\pi_p$, that represents the planner. Then the regularization term in the loss function is:

$$J(\theta) = \frac{1}{N} \sum_{i=0}^{N} \mathbb{E}_{a \sim \pi_\theta(\cdot|s)}[\log \pi_\theta(a|s) - \log \pi_p(a|s)], \tag{39}$$

with its gradient being:

$$\nabla J(\theta) = \nabla \frac{1}{N} \sum_{i=0}^{N} \mathbb{E}_{a \sim \pi_\theta(\cdot|s)}[\log \pi_\theta(a|s) - \log \pi_p(a|s)] \tag{40}$$

$$= \frac{1}{N} \sum_{i=0}^{N} \nabla \mathbb{E}_{a \sim \pi_\theta(\cdot|s)}[\log \pi_\theta(a|s)] - \frac{1}{N} \sum_{i=0}^{N} \nabla \mathbb{E}_{a \sim \pi_\theta(\cdot|s)}[\log \pi_p(a|s)]$$

We isolate and focus on the cross-entropy term since it is the only one that depends on $\pi_p$:

$$g(\theta) = \frac{1}{N} \sum_{i=0}^{N} \nabla \mathbb{E}_{a \sim \pi_\theta(\cdot|s)}[\log \pi_p(a|s)] = \frac{1}{N} \sum_{i=0}^{N} \mathbb{E}_{a \sim \pi_\theta(\cdot|s)}[\nabla \log \pi_\theta(a|s) \log \pi_p(a|s)] \tag{41}$$

## G.1. Expected value

**Previously stored planner policies.** Let $\pi^{k_i}$ be policy planner stored at time step $k_i \in \mathcal{K}$ where $i = 1, ..., N$, and let $k_i$ be sampled uniformly from $\mathcal{K}$. Substituting $\pi_p$ in Equation 41 yields:

$$g_1(\theta) = \mathbb{E}_{a \sim \pi_\theta(\cdot|s)}[\nabla_\theta \log \pi_\theta(a|s) (\frac{1}{N} \sum_{i=1}^{N} \log \pi^{k_i}(a|s))] \tag{42}$$

Next, the expected value of the gradient is:

$$\mathbb{E}_k[g_1(\theta)] = \mathbb{E}_{a \sim \pi_\theta(\cdot|s)}[\nabla_\theta \log \pi_\theta(a|s) \mathbb{E}_k[\log \pi^k(a|s)]] \tag{43}$$

**Learned policy prior.** Now let us develop the effect of learning first a prior from these samples, using it as an alternative to regularize the sampling policy $\pi_\theta$. Let $\pi_{\theta_p}$ be learned policy prior that iteratively minimizes the reverse KL divergence between itself and the sequence of previously stored planner policies $\pi^k$:

$$\pi_{\theta_p} := \arg\min_\pi \frac{1}{K} \sum_{k=0}^{K} \mathbb{E}_{a \sim \pi_{\theta_p}}[\log \pi_{\theta_p}(a|s) - \log \pi^k(a|s)] = \mathbb{E}_k[\log \pi^k(a|s)]. \tag{44}$$

Then it follows that $\log \pi_{\theta_p} = \mathbb{E}_k[\log \pi^k(a|s)]$. This is the first important result since substituting $\pi_p$ in Equation 41 now yields,

$$g_2(\theta) = \mathbb{E}_{a \sim \pi_\theta(\cdot|s)}[\nabla_\theta \log \pi_\theta(a|s) \log \pi_{\theta_p}(a|s)] = \mathbb{E}_{a \sim \pi_\theta(\cdot|s)}[\nabla_\theta \log \pi_\theta(a|s) \mathbb{E}_k[\log \pi^k(a|s)]], \tag{45}$$

Therefore, when using the learned policy prior, the mean of the gradient term that depends on $\pi_p$ is theoretically equivalent to using previously stored samples: $\mathbb{E}_k[g_1] = \mathbb{E}_k[g_2]$.

## G.2. Variance

**Previously stored planner policies.** The source of randomness for $g_1$ comes both from the sampled actions and the sampled planner policies. Thus, using the previous result, we can make the following decomposition:

$$\frac{1}{N}\sum_{i=1}^{N}\log\pi^{k_i}(a|s) = \mathbb{E}_k[\log\pi^k(a|s)] + \epsilon_N(a), \tag{46}$$

where $\mathbb{E}_k[\epsilon_N(a)] = 0$ and $\mathrm{Var}_k(\epsilon_N(a)) = \frac{1}{N}\mathrm{Var}_k(\log\pi^k(a|s))$

Then $g_1(\theta) = \mathbb{E}_{a\sim\pi_\theta(\cdot|s)}[\nabla_\theta\log\pi_\theta(a|s)\mathbb{E}_k[\log\pi^k(a|s)]] + \mathbb{E}_{a\sim\pi_\theta(\cdot|s)}[\nabla_\theta\log\pi_\theta(a|s)\epsilon_N(a)]$. Using the law of total variance and that $\epsilon_N(a)$ has zero mean we obtain:

$$\mathrm{Var}_{\substack{k\\a\sim\pi_\theta}}(g_1(a)) = \mathrm{Var}_{a\sim\pi_\theta}(\mathbb{E}_k[g_1|a]) + \mathbb{E}_{a\sim\pi_\theta}[\mathrm{Var}_k(g_1|a)] \tag{47}$$

$$= \mathrm{Var}_{a\sim\pi_\theta}(\nabla_\theta\log\pi_\theta(a|s)\mathbb{E}_k[\log\pi^k(a|s)])$$

$$+ \mathbb{E}_{a\sim\pi_\theta}\left[(\nabla_\theta\log\pi_\theta(a|s))(\nabla_\theta\log\pi_\theta(a|s))^T\frac{\mathrm{Var}_k(\log\pi^k(a|s))}{N}\right]$$

**Learned policy prior.** Assuming $\pi_{\theta_p}$ to be exact, the only source of randomness in $g_2$ is the sampling of actions $a$: $\mathrm{Var}_{a\sim\pi_\theta}(g_2) = \mathrm{Var}_{a\sim\pi_\theta}(\nabla_\theta\log\pi_\theta(a|s)\mathbb{E}_k[\log\pi^k(a|s)])$.

Which is identical to the first term in $\mathrm{Var}_{\substack{k\\a\sim\pi_\theta}}(g_1(a))$. Then it follows that, under the condition that $N \ll K$ and that $\pi^k$ are not all identical, the variance of gradient $g_2$ will be strictly lower than $g_1$.

