# OpenReview forum: "A KL-regularization framework for learning to plan with adaptive priors"
_ICML.cc/2026/Conference — ICML 2026 regular_

### Official Review · Reviewer_uELv · 2026-02-25

**Soundness:** 2
**Presentation:** 2
**Significance:** 2
**Originality:** 2
**Overall Recommendation:** 4
**Confidence:** 3

**Summary:**

This paper aims to address the mismatch between planning and policy learning in MPPI based RL algorithms, where the policy and value function are trained using data not from the planner samples and thus might not be useful or mislead planning. The proposed method is to add a KL penalty to keep the sampling policy close to planner distribution (via a learner planner prior that does behavior cloning of planner output data). Experiments are conducted on DM control and HumanoidBench using TD-MPC2 and BMPC as baselines and show enhanced performance.

**Compliance With Llm Reviewing Policy:**

Affirmed.

**Final Justification:**

I believe this paper warrants acceptance because it provides a unified lens and a sensible method over the mismatch problem in planning and policy learning.

My concern on the clarity of the paper's contribution is fully addressed, as the authors are committed to provide more context in the preliminaries.

I am raising my score to 4.

**Key Questions For Authors:**

My main question for the authors is to better explain the mismatch problem such that it's clear what the proposed solution is targeting and that the problem is worth addressing.

**Limitations:**

Yes

**Strengths And Weaknesses:**

**Strength**
* The paper addresses a potentially useful problem in MBRL & MPC
* The propose algorithm is relatively straightforward and the results show some advantage

**Weakness**
* The main issue with the paper is it's not super clear what the mismatch is. More explanation in the Preliminaries section would be helpful. I have an intuitive sense that the policy is not trained on MPC intermediate samples and thus can introduce value estimation errors or not providing good guide for MPC optimization. It will be good to add a simple toy example that mechanistically illustrate this problem and demonstrate the problem worthy of solving.

---

> ### Author Rebuttal · Authors · 2026-03-31
>
> We thank the reviewer for the time spent on the review and the useful comments. We address first the weaknesses, and afterwards reply to the questions one by one.
>
> - **Clarifications regarding the summary:** We want to clarify that the identification of the mismatch between planning and policy learning in MPPI-based RL algorithms is not our contribution. Instead, BMPC [2] is one of the first methods to identify this problem in TD-MPC2 [1] and provide a solution. We show that a principled interpolation is a better solution. Our novelty is exactly three things:
>   - A novel formulation of learning the sampling policy as KL-regularized RL at the trajectory level, hence the modifications in Eq. 7, recovering both known methods as limiting cases of the regularization strength.
>   - We demonstrate that intermediate values of $\lambda$ not only outperform both extremes but also manage to learn in environments where TD-MPC2 and BMPC fail to learn or solve the task, e.g. Balance Simple, Stair, Slide.
>   - We also show that the Learned policy prior reduces variance, both theoretically (Appendix G) and empirically (Figure 7), and study the effects of learning different policy priors.
>
>
> ## Weaknesses
> - **On the clarity of policy mismatch:** We appreciate the feedback from the reviewer. We did not include the challenge of policy mismatch between MPPI and sampling policy because it has been clearly explained before in the literature, which made it more convenient to refer the reader to [1]. We will include a subsection in preliminaries giving a summary of the challenge of policy mismatch and a small toy example.
>
> ## Questions
> 1. Policy mismatch refers to the mismatch of the behavior policy (the planner / MPPI) and the sampling policy. Assuming the learned bootstrap Q-value function is non-convex in the action space, policy mismatch arises and affects performance in the following manner:
>     - MPPI samples many sequences, some of them with the sampling policy, and evaluates them by estimating their H-step Q-value, bootstrapping at the H-step with the bootstrap Q-value function from the sampling policy. It chooses an action and interacts with the environment. The action selected is likely to have a better Q-value than the ones suggested by the sampling policy.
>     - The sampling policy in TD-MPC2 [1] is trained to locally maximize its bootstrap Q-value function. Unless constrained to remain close to the MPPI, it is unlikely for this local maximum to match or be close to the MPPI’s. This is the root of the policy mismatch, which causes the performance gap between the two policies.
>     - Since the bootstrap Q-value function assumes following the sampling policy, it will be updated with Q-targets evaluated in underexplored state-action couples. This is the effect of the policy mismatch and the cause of the MPPI policy performance drop.
>
>     This phenomenon has been previously identified in the literature. In particular, Wang, Y. et al. (2025) [2], Figure 2, shows an increasing evaluation performance gap between the MPPI policy and the sampling policy. This indicates inaccurate value estimation, further degrading the performance of MPPI.
>
> [1] Hansen, N., Su, H., & Wang, X. (2024). TD-MPC2: Scalable, robust world models for continuous control. In *Proceedings of the Twelfth International Conference on Learning Representations*.
>
> [2] Wang, Y., Guo, H., Wang, S., Qian, L., & Lan, X. (2025). Bootstrapped model predictive control. In *Proceedings of the Thirteenth International Conference on Learning Representations*

---

> > ### Author Rebuttal · Reviewer_uELv · 2026-04-04
> >
> > Thank the authors for the explanation. I think the problem setting is much clearer.
> >
> > I have a follow up question: looking at table 1, the proposed method which uses interpolation mainly outperforms TD-MPC on a subset of the environments, i.e., the environments between H1hand-hurdle and H1hand-stair. The performances on other environment are roughly on par. What characteristics of these environments contribute to the performance gap and the effectiveness of the proposed method? In other word, why the proposed method does not differentiate in other environments?

---

> > > ### Author Response · Authors · 2026-04-07
> > >
> > > We thank the reviewer for the positive feedback. We are glad that your earlier concern was resolved.
> > >
> > > Indeed, our method clearly outperforms TD-MPC2 in ~70% of the environments, BMPC in ~50% of the environments. In the other environments, our method still slightly outperforms or matches the performance of the baselines.
> > >
> > > Finding why our method does not differentiate as much in the latter environments is non-trivial. We hypothesize this might be due to two reasons:
> > > Exploration-hard environments (i.e. Maze, Balance-hard) require either high generalization or hierarchical policies. In any case, these environments require such a high degree of exploration that none of the methods manage to find a solution that solves the task.
> > > In simple environments, policy mismatch might not be a problem to begin with. In that case, the policy obtained by maximizing the Q-value function would converge to a similar solution to the planner.
> > >
> > > In summary, in hard exploration tasks, all methods fail, and in simpler tasks mismatch is minimal, explaining the smaller differences. Overall, our results highlight that properly balancing planner alignment and policy optimization is key, and that neither ignoring nor fully enforcing this alignment (as in TD-MPC2 and BMPC) is sufficient. We shall add this analysis in the reviewed version of the paper.
> > >
> > >  With this additional clarification, we hope to solve the reviewer’s concerns and also hope they would reconsider their score, given that all other concerns have been addressed as well.

---

### Official Review · Reviewer_4Moa · 2026-03-09

**Soundness:** 2
**Presentation:** 2
**Significance:** 2
**Originality:** 3
**Overall Recommendation:** 4
**Confidence:** 2

**Summary:**

This paper primarily discusses how to achieve effective exploration in model-based RL algorithms for high-dimensional continuous control tasks. In short, early methods based on model-predicted path integrals (i.e., MPPI) may suffer from mismatches between the policy and planner during training, which can negatively impact algorithm performance. Recent methods aim to alleviate this problem by minimizing the KL divergence between the planner and policy. This paper aims to implement a unified policy optimization and model predictive control framework (i.e., PO-MPC) that unifies these existing methods. The authors demonstrate that existing methods can be considered as special cases of the proposed framework and explore some novel variations. Extensive experiments are also conducted to demonstrate the effectiveness of the proposed methods.

**Compliance With Llm Reviewing Policy:**

Affirmed.

**Final Justification:**

The author's supplementary results and analysis have addressed my concerns; therefore, I am inclined to update my rating.

**Key Questions For Authors:**

See Weakness, and:

1. The authors could conduct more statistical significance analysis and try more random seeds to demonstrate that the proposed method does indeed have a significant improvement, rather than just random noise.

2. The sensitivity of hyperparameters and how to adaptively select them may be crucial for applying the proposed method to downstream tasks. I would appreciate it if the authors could discuss this further.

**Limitations:**

The authors discuss some limitations, but further discussion is still beneficial: 1) the statistical significance of the experiments; 2) the trade-off between the computational cost and performance of the additional components.

**Strengths And Weaknesses:**

## Strengths

Analyzing and using different model-based RL methods from a unified perspective is indeed interesting. Furthermore, the unified analysis of MPPI and existing KL-based methods clarifies their relationships, which is helpful to the community and helps readers better understand their relationships and development. Figure 1 is also good.

The experiments in this paper are extensive, covering multiple tasks. The improvements on some tasks are significant, providing some support for the proposed method.

The writing  of this paper is also good, and it is relatively easy to understand.

## Weaknesses

This paper benefits from using more random seeds in its experiments, thereby enhancing the persuasiveness and credibility of the experiments. Furthermore, a deeper discussion of the impact and sensitivity of different random seeds on the proposed algorithm might increase the paper's impact.

In addition, the sensitivity of some algorithm hyperparameters is worth analyzing. If the authors could discuss this further, such as under what settings the proposed method will fail, it could greatly benefit the community in using the proposed method.

Furthermore, if the authors could provide a more detailed analysis of computational costs, readers would better understand the differences between different algorithms, which could help in selecting the appropriate algorithm.

---

> ### Author Rebuttal · Authors · 2026-03-31
>
> We thank the reviewer for the time spent on the review and the useful comments. We address first the weaknesses, and afterwards reply to the questions one by one.
>
> ## Weaknesses
>
> - **On the use of more random seeds in our experiments**: We have extended the number of seeds in our results from 3 to 5. Since we now have 5 seeds, we have chosen to switch to practices from [1], replacing mean $\pm$ CI with IQM + bootstrap CI, with the aim of significantly improving the robustness and reliability of our comparisons.
>
>    The results are in the shared folder: https://drive.google.com/drive/folders/1h1TCk6lKupEd3hH6m2FQRkDgDGGDkxqr?usp=sharing
>
>     Where the last row is the aggregate IQM across tasks in each benchmark, with its corresponding stratified bootstrap CI. We want to stress that the additional results do not change the conclusions we obtained previously. We will include these results in the revised version of our work. The sensitivity of different random seeds on the proposed algorithm is shown through the aggregation of the multiple seeds we have trained, which is standard in RL.
> &nbsp;
>  - **On the analysis of the sensitivity of algorithm hyperparameters**: Our main contribution is not a single algorithm but a continuous design space for MPPI-based RL. Prior work corresponds to extreme points and one choice of policy prior ($\lambda=0$, $\lambda \longrightarrow \infty$ using planner samples as prior), and we show that intermediate regimes consistently outperform both. **Therefore, the only hyperparameters that our algorithm introduces over TD-MPC2 and BMPC are the hyperparameter $\lambda$ and the choice of prior $\pi_p$. This is shown in Figure 1, and Table 3 in appendix A.**  This is why we have done an ablation study that also shows empirically how different values of $\lambda$ affect evaluation performance throughout training. In that regard, we must remind that, apart from being baselines, TD-MPC2 and BMPC correspond to $\lambda$ values $\lambda=0$ and $\lambda \longrightarrow \infty$. In this context, we show that the framework’s performance degrades when $\lambda$ is too low ( $\lambda \leq 0.1$, policy mismatch), as well as in the limit $\lambda \longrightarrow \infty$ (BMPC, hinders exploration due to potential premature mode collapse), **as illustrated in Table 1, and Figures 4, and 5**. We conclude that λ=1 works well across most tasks, with larger λ often helping in high-dimensional settings. We further analyze how this sensitivity to $\lambda$ depends on the choice of policy prior $\pi_p$ used to constrain the trajectory distribution induced by the sampling policy, **as shown in Figures 2, and 3, and appendix D.3**.
>
>     We prioritised analysing the most relevant hyperparameters for the sake of compute constraints, since every seed takes almost one week of compute at best. We agree that further analysis of additional hyperparameters could be interesting. However, since these are not part of our main contribution, we consider such an investigation to be out of scope of the current paper.
> &nbsp;
> - **On the analysis of computational costs**: We thank the reviewer for the feedback. MPPI-based RL algorithms are complex frameworks, which makes it difficult to pinpoint the computational cost of each moving part. **It is easier to note the differences relative to TD-MPC2, especially in terms of the number of parameters.** At training time, our algorithm maintains the same structure as TD-MPC2, except for the extra KL-regularized Q-value function for updating the sampling policy, and, if included, the extra learned policy prior that clones the MPPI;**both changes are reported in Figure 1 as numbers 1 and 2**. In practice, we did not observe significant training duration changes between the baselines and our method. At inference time, our algorithm is computationally equivalent to TD-MPC2 since both policy prior and KL-regularized Q-value function are no longer needed. We will add this information in the revised version of the paper, within the first paragraph of the experiments section (Section 5).
>
> ## Questions
>
> 1. Please see our answer to *the first point in Weaknesses*.
> 2. Please see our answer to *the second point in Weaknesses*.
>
> Finally, we also want to add that our limitations section is quite complete, and **the trade-off between the computational cost and performance of the additional components is actually tested and discussed**. First, we compare and discuss POMPC against TD-MPC2 in Table 1 and Figures 4 and 5. Second, we compare and discuss POMPC trained with a learned policy prior against POMPC trained with a policy prior extracted from replay buffer samples in Figures 2 and 6.

---

> > ### Author Rebuttal · Reviewer_4Moa · 2026-04-02
> >
> > I appreciate the author's supplementary results and analysis, which have addressed my concerns; therefore, I am inclined to update my rating.

---

> > > ### Author Response · Authors · 2026-04-03
> > >
> > > We sincerely thank you for your constructive review. Please let us know if you have any further concerns.

---

### Official Review · Reviewer_wptB · 2026-03-09

**Soundness:** 3
**Presentation:** 3
**Significance:** 2
**Originality:** 2
**Overall Recommendation:** 4
**Confidence:** 3

**Summary:**

Addressing the challenge of exploration efficiency in high-dimensional continuous control tasks within model-based reinforcement learning (MBRL), this paper proposes a KL-regularized framework named PO-MPC (Policy Optimization–Model Predictive Control), which formulates sampling policy learning as a KL-regularized RL problem that takes the action distribution of the MPPI planner as the prior. This framework resolves the issues of distribution mismatch between the sampling policy and the planner, as well as the variance induced by outdated planning samples. In this study, an adaptive prior is introduced, and investigations are conducted into different configurations of the KL regularization strength λ and various loss functions for prior training. Experimental results on 7 tasks from the DeepMind Control Suite and 14 tasks from the HumanoidBench benchmark demonstrate that PO-MPC achieves significant improvements in both sample efficiency and final performance compared with state-of-the-art (SOTA) methods such as TD-MPC2 and BMPC. Meanwhile, this framework unifies existing MPPI-based RL methods as its special cases.

**Compliance With Llm Reviewing Policy:**

Affirmed.

**Final Justification:**

The authors' clarifications and additional experimental results have addressed my remaining concerns satisfactorily. We have adjusted our score to weak accept.

**Key Questions For Authors:**

1. Could additional analysis be provided on the correlation between λ and α, along with ablation experiments investigating such a correlation? It would also be valuable to present the performance of the algorithm with the α hyperparameter removed.
2. Could more state-of-the-art MPPI-based and relevant baseline algorithms be included for comparative analysis?
3. Could the consistency of the results in the figures and tables be verified? In addition, could the mean value be calculated over multiple random seeds to eliminate the impact of randomness in the experimental results?
4. Furthermore, is it possible to adaptively adjust the λ hyperparameter according to the training progress and the characteristics of the sampled data?

**Limitations:**

yes

**Strengths And Weaknesses:**

Strengths
1. The research motivation of the paper is well-defined, with a complete structure and clear presentation.
2. Necessary theoretical analyses are provided (yet the derivations remain unfinished, and the validity of the theories is to be verified).


Weaknesses
1. The paper appears to merely combine the core ideas of the TD-MPC2 and BMPC algorithms, leading to a lack of sufficient novelty in its contributions.
2. The paper constrains the deviation between the sampling policy and the planning policy via the KL regularization term and the information entropy regularization term, while preserving the exploration capability of the algorithm. The balance between the λ and α hyperparameters thus seems to exert a critical influence on the convergence characteristics of the algorithm. However, the paper fails to conduct an in-depth analysis of the correlation between λ and α, nor does it present experimental investigations into such a correlation. Only ablation experiments on the single hyperparameter λ are provided, which is insufficient to elaborate on the relationship between the two hyperparameters and their selection criteria.
3. Although the paper refers to multiple MPPI-based algorithms, the only baseline algorithms adopted in the experiments are BMPC and TD-MPC2. It would be more convincing to incorporate more state-of-the-art baseline algorithms for comparative analysis.
4. In addition, the results presented in Figure 4 of the appendix seem to indicate that the BMPC algorithm achieves better training performance, and the error bands suggest a smaller variance in its results. Nevertheless, the results shown in the tables draw the exact opposite conclusion, creating an inconsistency that requires clarification.

---

> ### Author Rebuttal · Authors · 2026-03-31
>
> We thank the reviewer for the time spent on the review and the useful comments. We address the weaknesses and reply to the questions one by one.
>
> ## Weaknesses
>
> - **On the lack of sufficient novelty due to the method being a mere combination of the core ideas of TDMPC2 and BMPC**: We respectfully disagree with this statement. Within MPPI-based RL, previous work either learn sampling policies that **maximize the Q-value** (TD-MPC2) or **clones the planner** through reverse KL minimization (BMPC). We show that a principled interpolation is a better solution. We kindly refer the reviewer to our response to Reviewer uELv for further details on our novelty.
>
>  - **On the use of entropy regularization and the interplay of $\lambda$ and $\alpha$**: We believe the concern about the interaction between λ and α may stem from a misunderstanding of their respective roles. Entropy regularization (α) is applied only during the policy update and primarily increases the variance of the Gaussian sampling policy. This is a standard component in actor-critic methods (e.g., PPO, TD-MPC2, BMPC) and is not part of our contribution; accordingly, we fix α to match the baselines (α = 1e−4).
> Our contribution instead focuses on the KL regularization controlled by λ. We provide an empirical analysis over λ ∈ {0, 0.1, 1, 9} as well as in the limit λ → ∞. Notably, the baselines correspond to extreme cases of λ, so these ablations already capture the full range of behaviors induced by this parameter within our framework.
> Given the standard and fixed role of α, we believe the presented analysis sufficiently characterizes the effect of λ. As noted in the limitations section, we recommend λ = 1 as a practical default, equally balancing KL regularization and Q-value maximization.
>
> - **On the addition of baselines to the paper**: We want to clarify that most of these MPPI algorithms are still unpublished pre-prints, while BMPC and TD-MPC2 are very recent and two of the strongest baselines in the state-of-the-art. The number of baselines we employ does not deviate from the norm, and BMPC and TD-MPC2 have been shown to dominate DreamerV3 and SAC over DMControl and HumanoidBench benchmarks, so adding them would not have changed the conclusions. Therefore, we prioritized the most relevant baselines for the sake of compute constraints.
>  In the revised version of our paper, we will add a comparison with the results presented in BOOM [1], another recently published method in the field of MPPI-based RL on a subset of locomotion tasks in HumanoidBench. Here is a table comparing our results with those of BOOM for 1M time steps. Our method still performs better, at best, than the state of the art and comparable, at worst.
>
> The results are in the shared folder: https://drive.google.com/drive/folders/1h1TCk6lKupEd3hH6m2FQRkDgDGGDkxqr?usp=sharing
>
> - **On the inconsistency of the results**: We believe there may be a misunderstanding. First, the results in Table 1 are obtained from the final performance of the same runs used to generate the learning curves in Figure 4 of Appendix D. Therefore, the results reported in Table 1 are consistent with those curves. In Figure 4 **our method clearly outperforms BMPC** in *humanoid-run* and *dog-stand*. We understand that the other plots are more difficult to visualize, hence we present the results in a table. Regarding variance, Figure 4 shows comparable variance for $\lambda=1$ or $\lambda=9$, depending on the environment.
>
>     We also understand that in lower-dimensional environments (DMControl), the performance difference is small, and difficult to visualize graphically. This first batch of results serves to show that our method improves OR, at least does not harm performance. However, the improvement in performance of our method is clearly visible in environments with a higher-dimensional action-space (HumanoidBench has almost 2x action dim. with respect to the dog environments from DMControl). **Our claim is confirmed by the other reviewers, who point out that “The proposed fix makes sense and seems to work in practice”, “The improvements on some tasks are significant, providing some support for the proposed method”, and “the results show some advantage”.**
>
> ## Questions
>
> 1. Please see our answer to the *second point in weaknesses* and our note on the computation cost of additional results in the *third point in weaknesses*.
> 2. Please see our answer to the *third point in weaknesses*
> 3. Please see our answer to the *fourth point in weaknesses*. Regarding the use of multiple seeds, please see our answer to Reviewer 4Moa.
> 4. This is indeed possible, and **we discuss it as the first item in our limitations**.  However, studying how to tune it automatically is out-of-scope for this paper; therefore left for future work.
>
>
> [1] Zhan, G., et al. (2025). Bootstrap off-policy with world model. In *Proceedings of the Thirty-Ninth Annual Conference on Neural Information Processing Systems*.

---

> > ### Author Rebuttal · Reviewer_wptB · 2026-04-04
> >
> > We thank the authors for their patient and detailed response. Most of the concerns I raised have been properly addressed. However, I still remain worried about the consistency of the experimental results presented in the paper. For this reason, I will keep my original score for the time being.

---

> > > ### Author Response · Authors · 2026-04-07
> > >
> > > We thank the reviewer for their time in reading our rebuttal, and we are glad that most of their concerns have been properly addressed. Regarding the remaining concern on the consistency of the results not being resolved, we respectfully disagree. This has actually been explicitly addressed in our fourth point: *On the inconsistency of the results*, and in our answer to Question 3.
> > >
> > > For completeness, we have added to the following shared folder: https://www.swisstransfer.com/d/155c21ea-cc36-40eb-b3c4-cca5ee85083c
> > > - All updated tables with 5 seeds
> > > - All updated Figures with 5 seeds
> > >
> > > Furthermore, we have added to our anonymous GitHub repository, given in the paper, under a folder called “paper_results”:
> > > - The pickle files with the data for each environment and seeds
> > > - The code we have used to load the data and produce the plots and tables.
> > >
> > > With this additional clarification, we hope to solve the reviewer’s concerns and also hope they would reconsider their score, given that all other concerns have been addressed as well.

---

### Official Review · Reviewer_vmU1 · 2026-03-20

**Soundness:** 3
**Presentation:** 3
**Significance:** 3
**Originality:** 3
**Overall Recommendation:** 5
**Confidence:** 3

**Summary:**

Current methods such as TD-MPC2 and BMPC attempt to solve hard continuous control problems by blending elements of planning and reinforcement learning.

For planning you need a model of the transition and of the reward that are learned from data. Full horizon planning is costly so a first approximation is to plan over short horizon H but still take into account the total task horizon T, by bootstrapping beyond H using a learned Q value for that planner.

Given the world model and Q, approximate planning is often made more efficient when “guided” by an extra policy pi that proposes good candidate sequences.

These extra RL elements only make sense if they remain aligned to the planner as the learning happens, the planner, Q and pi need to be aligned along the training.

TD-MPC2 uses these elements.
A Q-value is learned from planning data, but using the guiding policy pi in the target, hence it is a Q value of the guiding policy pi. To be a useful boostrap for planning on a short horizon, this guiding policy must be close to the planner.
In TD-MPC2, this guiding policy pi is learned via policy optimization maximizing that Qvalue.

The author identify the following problem:
For TD-MPC2 to work, the planner, Q and the guiding policy pi, must stay close (otherwise, the objectives used no longer make sense or are biased)
Yet, in TD-MPC2, nothing forces pi to stay close to the planner. Thus Q also diverges from the value of the planner.

Their proposal is to add a KL penalty (KL[ . | distilled planner]) to the objective for the guiding policy pi.

The authors empirically  demonstrate their method stabilize the learning on challenging problems and performs better than alternative methods

**Compliance With Llm Reviewing Policy:**

Affirmed.

**Key Questions For Authors:**

The authors propose a patch to TD-MPC2 which is fine.
Improvement in this method comes from 1) the guiding policy trained via RL, 2) the planner exploiting the model
In BMPC, improvement only comes from the planner.

Adding the RL part to learn the guiding comes at the cost of many approximations. Is it really worth it? BMPC is simpler but as a result maybe more stable? but does the gain in stability compensate for the slower improvement of the method?

Also, your patch slows the improvement (pi_s is more constrained), how does that fact factor in the debate?

**Strengths And Weaknesses:**

My review focuses on the method, I did not check the empirical evaluation in detail

# Strengths
* The authors identify a pathology of TD-MPC2 and explain it well
* The proposed fix makes sense and seems to work in practice
* the resulting objectives make sense

## Weakness
* more of a comment but I find the background section assumes a lot of knowledge (I don't have) and could be more self contained.

---

> ### Author Rebuttal · Authors · 2026-03-31
>
> We thank the reviewer for the time spent on the review and the useful comments. We will reply to the questions one by one.
>
> ## Questions
> 1. Learning the sampling policy through KL-regularized RL is indeed worth it. This is seen in high-dimensional environments from HumanoidBench (e.g. Table 1 and Figure 5), where both exploration and trajectory distribution constraints are necessary to optimize performance. We later show that all further modifications improve performance in Figures 2 and 3.
> 2. We agree that BMPC is simpler, but it ignores the information that can be obtained from maximizing the Q-value function.
> 3. We have tested the limits of our method and shown its robustness to different values of $\lambda$ in Table 1, and Figures 4 and 5. We show that it is stable and that it learns faster than the baselines in Figure 5.
> 4. Our framework is more constrained than TD-MPC2, but it is less constrained than BMPC, since our framework trades-off KL minimization of trajectory distributions with return maximization. Note that constraining the trajectory distribution induced by the sampling policy does not necessarily mean decreasing the speed of improvement. It means that the sampling policy will remain in the neighborhood of the planner’s distribution, reducing policy mismatch and decreasing the likelihood of evaluating the bootstrap Q-value function at unexplored state-action couples.

---

> > ### Author Rebuttal · Reviewer_vmU1 · 2026-04-04
> >
> > I have read the authors response and the other reviews and discussion.
> > My comments have been satisfyingly addressed

---

> > > ### Author Response · Authors · 2026-04-07
> > >
> > > We sincerely thank you for the positive feedback and your constructive review. Please let us know if you have any further concerns.

---

### Decision · Program_Chairs · 2026-04-30

**Decision:**

Accept (regular)

**Comment:**

This paper proposes a KL-regularized framework for MPPI-based model-based RL that unifies TD-MPC2 and BMPC as limiting cases and introduces adaptive planner priors for policy learning. Reviewers agreed that the paper addresses a relevant problem and that the proposed formulation is technically sound. The empirical results were generally viewed positively, especially on the higher-dimensional HumanoidBench tasks, where intermediate regularization strengths often improve over the main baselines. The paper also motivates the use of a learned intermediate prior to reduce variance, and this aspect is supported both empirically and theoretically.

The discussion was useful and three reviewers raised their scores after the rebuttal. The main points resolved were clarification of the consistency of the reported results and explanation of the policy-planner mismatch, novelty and the conditions where the method is effective.

In a revised version, the authors should incorporate the following points clarified during the rebuttal:
- A clearer explanation of planner-policy mismatch and the novelty.
- The results using multiple seeds and corresponding statistical evaluations, as well as clarification of the consistency between Table 1 and Figures 4 and 5. Also, a brief discussion of the task regimes in which the method helps most, practical guidance on the choice of $\lambda$ and a concise account of computational overhead.
- Importantly, a comparison to BOOM which has been proposed recently should also be added if feasible, as promised during discussion, at least on the overlapping benchmarks if a full comparison is not feasible.